# ANALYZING THE LATENT SPACE OF GAN THROUGH LOCAL DIMENSION ESTIMATION

## ABSTRACT

The impressive success of style-based GANs (StyleGANs) in high-fidelity image synthesis has motivated research to understand the semantic properties of their latent spaces. Recently, a close relationship was observed between the semantically disentangled local perturbations and the local PCA components in $\mathcal{W}$-space. However, understanding the number of disentangled perturbations remains challenging. Building upon this observation, we propose a local dimension estimation algorithm for an arbitrary intermediate layer in a pre-trained GAN model. The estimated intrinsic dimension corresponds to the number of disentangled local perturbations. In this perspective, we analyze the intermediate layers of the mapping network in StyleGANs. Our analysis clarifies the success of $\mathcal{W}$-space in StyleGAN and suggests a method for finding an alternative. Moreover, the intrinsic dimension estimation opens the possibility of unsupervised evaluation of global-basis-compatibility and disentanglement for a latent space. Our proposed metric, called *Distortion*, measures an inconsistency of intrinsic tangent space on the learned latent space. The metric is purely geometric and does not require any additional attribute information. Nevertheless, the metric shows a high correlation with the global-basis-compatibility and supervised disentanglement score. Our work is the first step towards selecting the most disentangled latent space among various latent spaces in a GAN without attribute labels.

## 1 INTRODUCTION

Generative Adversarial Networks (GANs) (Goodfellow et al., 2014) have achieved remarkable success in generating realistic high-resolution images (Karras et al., 2018; 2019; 2020b; 2021; 2020a; Brock et al., 2018). Nevertheless, understanding how GAN models represent the semantics of images in their latent spaces is still a challenging problem. To this end, several recent works investigated the disentanglement (Bengio et al., 2013) properties of the latent space in GAN (Goetschalckx et al., 2019; Jahanian et al., 2019; Plumerault et al., 2020; Shen et al., 2020). In this work, we concentrate on finding a disentangled latent space in a pre-trained model. A latent space is called (globally) disentangled if there is a bijective correspondence between each semantic attribute and each axis of latent space when represented with the optimal basis. (See the appendix for detail.)

The style-based GAN models (Karras et al., 2019; 2020b) have been popular in previous studies for identifying a disentangled latent space in a pre-trained model. First, the space of style vector, called $\mathcal{W}$-space, was shown to provide a better disentanglement property compared to the latent noise space $\mathcal{Z}$ (Karras et al., 2019). After that, several attempts have been made to discover other disentangled latent spaces, such as $\mathcal{W}^+$-space (Abdal et al., 2019) and $\mathcal{S}$-space (Wu et al., 2020). However, their better disentanglement was assessed by the manual inspection (Karras et al., 2019; Abdal et al., 2019; Wu et al., 2020) or by the quantitative scores employing a pre-trained feature extractor (PPL (Karras et al., 2019)) or an attribute annotator (Separability (Karras et al., 2019) and DCI metric (Eastwood & Williams, 2018; Wu et al., 2020)). The manual inspection is vulnerable to sample dependency, and the quantitative scores depend on the pre-trained models and the set of selected target attributes. Therefore, we need an *unsupervised* quantitative evaluation scheme for the disentanglement of latent space that *does not rely on pre-trained models*.

In this paper, we investigate the semantic property of a latent space by analyzing its geometrical property. In this regard, we propose a local intrinsic dimension estimation scheme for a learned

intermediate latent space in pre-trained GAN models. The local intrinsic dimension is the number of dimensions required to properly approximate the latent space locally (Fig 1a). We discover this intrinsic dimension by estimating the robust rank of Jacobian of the subnetwork. The estimated dimension is interpreted as the number of disentangled local perturbations. Furthermore, the intrinsic dimension of latent manifold leads to an unsupervised quantitative score for the global disentanglement property. The experiments demonstrate that our proposed metric shows a high correlation with the global-basis-compatibility and supervised disentanglement score. (The global-basis-compatibility will be rigorously defined in Sec 4.) Our contributions are as follows:

1. We propose a local intrinsic dimension estimation scheme for an intermediate latent space in pre-trained GAN models. The scheme is derived from the rank estimation algorithm applied to the Jacobian matrix of a subnetwork.

2. We propose a layer-wise global disentanglement score, called *Distortion*, that measures the inconsistency of intrinsic tangent space. The proposed metric shows a high correlation with the global-basis-compatibility and supervised disentanglement score.

3. We analyze the intermediate layers of the mapping network through the proposed Distortion metric. Our analysis elucidates the superior disentanglement of $\mathcal{W}$-space compared to the other intermediate layers and suggests a criterion for finding a similar-or-better alternative.

## 2 RELATED WORKS

**Style-based Generator**    Recently, GANs with style-based generator architecture (Karras et al., 2019; 2020b; 2021; Sauer et al., 2022) have achieved state-of-the-art performance in realistic image generation. In conventional GAN architecture, such as DCGAN (Radford et al., 2016) and ProGAN (Karras et al., 2018), the generator synthesizes an image by transforming a latent noise with a sequence of convolutional layers. On the other hand, the style-based generator consists of two subnetworks: *mapping network* $f : \mathcal{Z} \rightarrow \mathcal{W}$ and *synthesis network* $g : \mathbb{R}^{n_0} \times \mathcal{W}^L \rightarrow \mathcal{X}$. The synthesis network is similar to conventional generators in that it is composed of a series of convolutional layers $\{g_i\}_{i=1,\cdots,L}$. The key difference is that the synthesis network takes the learned constant feature $\mathbf{y}_0 \in \mathbb{R}^{n_0}$ at the first layer $g_0$, and then adjusts the output image by injecting the layer-wise styles $\mathbf{w}$ and noise (Layer-wise noise is omitted for brevity.):

$$\mathbf{y}_i = g_i(\mathbf{y}_{i-1}, \mathbf{w}) \quad \text{with } \mathbf{w} = f(\mathbf{z}) \quad \text{for } i = 1, \cdots, L, \tag{1}$$

where the style vector $\mathbf{w}$ is attained by transforming a latent noise $\mathbf{z}$ via the mapping network $f$.

**Understanding Latent Semantics.**    The previous attempts to understand the semantic property of latent spaces in StyleGANs are categorized into two topics: (i) finding more disentangled latent space in a model; (ii) discovering meaningful perturbation directions in a latent space corresponding to disentangled semantics. Several studies on (i) suggested various disentangled latent spaces in StyleGAN models, for example, $\mathcal{W}$ (Karras et al., 2019), $\mathcal{W}^+$ (Abdal et al., 2019), $P_N$ (Zhu et al., 2020), and $\mathcal{S}$-space (Wu et al., 2020). However, the superiority of the newly proposed latent space was demonstrated only through comparison with the previous latent spaces, not by selecting the best one among all candidates. Moreover, the comparison was conducted by manual inspections (Karras et al., 2019; Abdal et al., 2019; Wu et al., 2020) or by quantitative metrics relying on pre-trained models (Karras et al., 2019; Wu et al., 2020). Also, the previous works on (ii) are classified into local and global methods. The local methods find sample-wise perturbation directions (Ramesh et al., 2018; Patashnik et al., 2021; Abdal et al., 2021; Zhu et al., 2021; Choi et al., 2022b). On the other hand, the global methods search layer-wise perturbation directions that perform the same semantic manipulation on the entire latent space (Härkönen et al., 2020; Shen & Zhou, 2021; Voynov & Babenko, 2020). Throughout this paper, we refer to these local methods as *local basis* and these global methods as *global basis*. GANSpace (Härkönen et al., 2020) showed that the principal components obtained by PCA can serve as the global basis. SeFa (Shen & Zhou, 2021) suggested the singular vectors of the first weight parameter applied to latent noise as the global basis. These global basis showed promising results, but they were successful in a limited area. Depending on the sampled latent variables, these methods exhibited limited semantic factorization and sharp degradation of image fidelity Choi et al. (2022b;a). In this regard, (Choi et al., 2022b) suggested the need for diagnosing a global-basis-compatibility of latent space. Here, the global-basis-compatibility means how well the optimal global basis can work on the target latent space.

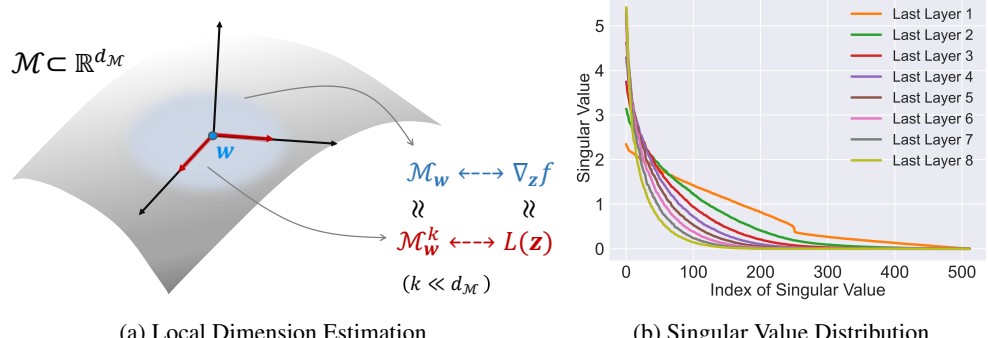

(a) Local Dimension Estimation

(b) Singular Value Distribution

Figure 1: **(a) Overview of Local Dimension Estimation.** Our goal is to find dimension $k$ such that the $k$-dimensional submanifold $\mathcal{M}_{\mathbf{w}}^k$ can properly describe the local latent manifold $\mathcal{M}_{\mathbf{w}} \subseteq \mathbb{R}^{d_{\mathcal{M}}}$. **(b) Singular Value Distribution** of Jacobian matrix for each subnetwork of the mapping network in StyleGAN2. As the layer gets deeper, many of the singular values are close to zero. This supports our claim of pruning latent dimension by interpreting the near-zero singular values as noise.

**Local Basis** Choi et al. (2022b) proposed an unsupervised method for finding local semantic perturbations based on the local geometry, called *Local Basis (LB)*. Throughout this paper, we denote Local Basis as LB to avoid confusion with the general term "local basis" in the previous paragraph. Assume the support $\mathcal{Z}$ of input prior distribution $p(\mathbf{z})$ is the entire Euclidean space, i.e., $\mathcal{Z} = \mathbb{R}^{d_{\mathcal{Z}}}$, for example, Gaussian prior $p(\mathbf{z}) = \mathcal{N}(0, I)$. We denote the target latent space by $\mathcal{M} = f(\mathcal{Z}) \subseteq \mathbb{R}^{d_{\mathcal{M}}}$ and refer to the subnetwork between them by $f$. Note that the target latent space $\mathcal{M}$ is defined as an image of the trained subnetwork $f$. Hence, we call $\mathcal{M}$ the *learned latent space* or the *learned latent manifold* following the manifold interpretation of Choi et al. (2022b).

LB is defined as the ordered basis of tangent space $T_{\mathbf{w}}\mathcal{M}_{\mathbf{w}}^k$ at $\mathbf{w} = f(\mathbf{z}) \in \mathcal{M}$ for the $k$-dimensional local approximating manifold $\mathcal{M}_{\mathbf{w}}^k$. Here, $\mathcal{M}_{\mathbf{w}}^k \subseteq \mathcal{M}$ indicates a $k$-dimensional submanifold of $\mathcal{M}$ that approximates $\mathcal{M}$ around $\mathbf{w}$ (Fig 1a) with $k \leq d_{\mathcal{M}}$:

$$\mathcal{M}_{\mathbf{w}}^k \approx \mathcal{M}_{\mathbf{w}} \quad \text{where} \quad \mathcal{M}_{\mathbf{w}} = \{f(\mathbf{z}_\epsilon) \mid \|\mathbf{z}_\epsilon - \mathbf{z}\| < \epsilon\} \subseteq \mathcal{M}. \tag{2}$$

Using the fact that $\mathcal{M} = f(\mathcal{Z})$, the local approximating manifold $\mathcal{M}_{\mathbf{w}}^k$ can be discovered by solving the low-rank approximation problem of $df_{\mathbf{z}}$, i.e., the Jacobian matrix $\nabla_{\mathbf{z}} f$ of $f$:

$$\text{minimize}_L \quad \|df_{\mathbf{z}} - L\|_2, \qquad \text{where } \text{rank}(L) \leq k. \tag{3}$$

The analytic solution of this low-rank approximation problem is obtained in terms of Singular Value Decomposition (SVD) by Eckart–Young–Mirsky Theorem (Eckart & Young, 1936). From that, $\mathcal{M}_{\mathbf{w}}^k$ and the corresponding LB are given as follows: For the $i$-th singular vector $\mathbf{u}_i^{\mathbf{z}} \in \mathbb{R}^{d_{\mathcal{Z}}}$, $\mathbf{v}_i^{\mathbf{w}} \in \mathbb{R}^{d_{\mathcal{M}}}$, and $i$-th singular value $\sigma_i^{\mathbf{z}} \in \mathbb{R}$ of $df_{\mathbf{z}}$ with $\sigma_1^{\mathbf{z}} \geq \cdots \geq \sigma_n^{\mathbf{z}}$,

$$df_{\mathbf{z}}(\mathbf{u}_i^{\mathbf{z}}) = \sigma_i^{\mathbf{z}} \cdot \mathbf{v}_i^{\mathbf{w}} \quad \text{for } \forall i, \qquad \text{LB}(\mathbf{w} = f(\mathbf{z})) = \{\mathbf{v}_i^{\mathbf{w}}\}_{1 \leq i \leq n}, \tag{4}$$

$$\mathcal{M}_{\mathbf{w}}^k = \left\{ f\left(\mathbf{z} + \sum_i t_i \cdot \mathbf{u}_i^{\mathbf{z}}\right) \mid t_i \in (-\epsilon_i, \epsilon_i), \text{ for } 1 \leq i \leq k \right\}. \tag{5}$$

Note that the tangent space of $\mathcal{M}_{\mathbf{w}}^k$ is spanned by the top-$k$ LB, i.e. $T_{\mathbf{w}}\mathcal{M}_{\mathbf{w}}^k = \text{span}\{\mathbf{v}_i^{\mathbf{w}} : 1 \leq i \leq k\}$. Therefore, traversing along LB is guaranteed to stay close to the latent manifold, thereby providing a strong robustness of image quality. However, Choi et al. (2022b) did not provide an estimate on the number of meaningful perturbations. Since LB is defined as singular vectors, Choi et al. (2022b) presents the candidates as much as the ambient dimension. In this regard, we propose the local dimension estimation that can refine these candidates up to 90%. Moreover, this local dimension estimation leads to an unsupervised global disentanglement metric (Sec 4).

## 3 LATENT DIMENSION ESTIMATION

In this section, we propose a local dimension estimation scheme for a *learned* latent manifold in a pre-trained GAN model. Following the work of Choi et al. (2022b), the estimated local dimension at

$\mathbf{w} \in \mathcal{M}$ corresponds to the number of local semantic perturbations from $\mathbf{w}$. The proposed scheme is based on the rank estimation algorithm (Kritchman & Nadler, 2008) applied to the differential of subnetwork $df_{\mathbf{z}}$. Then, we evaluate the validity of the estimated local dimension. In this section, our analysis of learned latent manifold is focused on the intermediate layers in the mapping network of StyleGAN2 (Karras et al., 2020b) trained on FFHQ (Karras et al., 2019). However, the proposed scheme can be applied to any $\mathbf{z}$-differentiable intermediate layers for an input latent noise $\mathbf{z}$.

### 3.1 METHOD

Throughout this work, we follow the notation presented in Sec 2. Consider a target latent space $\mathcal{M}$ given by a subnetwork $f$, i.e., $\mathcal{M} = f(\mathcal{Z})$. Our goal is to estimate the intrinsic local dimension of the learned latent manifold $\mathcal{M}$ around $\mathbf{w} = f(\mathbf{z})$. Geometrically, this intrinsic dimension represents the dimension required to locally describe the major variations of the manifold. The intrinsic local dimension is discovered by interpreting the differential $df_{\mathbf{z}}$ as a noisy linear map and finding its intrinsic rank. The correspondence between the local dimension and rank of $df_{\mathbf{z}}$ is described in Eq 5 because the rank of a linear map is the same as the number of singular vectors with non-zero singular values. Note that the matrix representation of $df_{\mathbf{z}}$ is a Jacobian matrix $(\nabla_{\mathbf{z}} f)(\mathbf{z})$.

**Motivation**   Before presenting our dimension estimation algorithm, we provide motivation for introducing the lower-dimensional approximation to $\mathcal{M}$. Figure 1b shows the singular value distribution of Jacobian matrices evaluated for the subnetworks of the mapping network in StyleGAN2. *Last layer $i$* in Fig 1b denotes the subnetwork from the input noise space $\mathcal{Z}$ to the $i$-th fully connected layer. The distribution of singular values $\{\sigma_i^{\mathbf{z}}\}_i$ gets monotonically sparser as the subnetwork gets deeper. In particular, $\mathcal{W}$-space, i.e., *Last layer 8*, is extremely sparse as much as $\sigma_{150}^{\mathbf{z}}/\sigma_1^{\mathbf{z}} \approx 0.005$. Therefore, it is reasonable to prune the singular values with negligible magnitude and consider the lower-dimensional approximation of the learned latent manifold.

**Pseudorank Algorithm**   The intrinsic rank estimation algorithm distinguishes the large meaningful components and the small noise-like components given the singular values $\{\sigma_i^{\mathbf{z}}\}_i$ of $(\nabla_{\mathbf{z}} f)(\mathbf{z})$. The Pseudorank algorithm (Kritchman & Nadler, 2008) determines the number of meaningful components based on the theoretical results from the random matrix theory literature. Assume the *isotropic Gaussian noise* on the Jacobian $(\nabla_{\mathbf{z}} f)(\mathbf{z}) \in \mathbb{R}^{d_{\mathcal{M}} \times d_{\mathcal{Z}}}$:

$$(\nabla_{\mathbf{z}} f)(\mathbf{z}) = L(\mathbf{z}) + \sigma \cdot (\epsilon_1, \cdots, \epsilon_{d_{\mathcal{M}}})^{\mathsf{T}} \quad \text{with} \quad \epsilon_i \sim \mathcal{N}(0, I_{d_{\mathcal{Z}}}), \tag{6}$$

where $L(\mathbf{z})$ denotes the denoised low-rank representation of $(\nabla_{\mathbf{z}} f)(\mathbf{z})$. Then, taking the expectation over the noise distribution gives:

$$\mathbb{E}_{\epsilon} \left[ (\nabla_{\mathbf{z}} f)^{\mathsf{T}}(\mathbf{z}) \cdot (\nabla_{\mathbf{z}} f)(\mathbf{z}) \right] = L^{\mathsf{T}}(\mathbf{z}) \cdot L(\mathbf{z}) + \sigma^2 \cdot I_{d_{\mathcal{Z}}}. \tag{7}$$

The eigenvalues of $[(\nabla_{\mathbf{z}} f)^{\mathsf{T}}(\mathbf{z}) \cdot (\nabla_{\mathbf{z}} f)(\mathbf{z})]$ are the squares of sigular values $\{(\sigma_i^{\mathbf{z}})^2\}_i$, and the noise covariance term $\sigma^2 \cdot I_{d_{\mathcal{Z}}}$ increases all eigenvalues by $\sigma^2$. This observation explains our intuition that large singular values correspond to signals and small ones correspond to noise. Therefore, determining the intrinsic rank of $(\nabla_{\mathbf{z}}) f(\mathbf{z})$ is closely related to the largest eigenvalue $\lambda_1$ of the empirical covariance matrix $S = \frac{1}{d_{\mathcal{Z}}} \sum_i \epsilon_i \cdot \epsilon_i^{\mathsf{T}}$, which is the threshold for distinguishing between signal and noise. The Pseudorank algorithm is based on the theoretical results of the asymptotic behavior of $\lambda_1$. The distribution of the largest eigenvalue $\lambda_1$ of the empirical covariance matrix for $n$-samples of $\mathcal{N}(0, I_p)$ converges to a Tracy-Widom distribution $F_{\beta}$ of order $\beta = 1$ for real-valued observations (Johnstone, 2001) (See the appendix for detail.):

$$P \left( \lambda_1 < \sigma^2 \left( \mu_{n,p} + s \cdot \sigma_{n,p} \right) \right) \to F_{\beta}(s) \quad \text{as } n, p \to \infty \text{ with } c = p/n \text{ fixed}. \tag{8}$$

Here, note that we do not know the true noise level $\sigma$ in Eq 6 a priori. Using the above theoretical results, the Pseudorank algorithm applies a sequence of nested hypothesis tests. Given the Jacobian $(\nabla_{\mathbf{z}} f)(\mathbf{z}) \in \mathbb{R}^{d_{\mathcal{M}} \times d_{\mathcal{Z}}}$ and let $p = \min(d_{\mathcal{M}}, d_{\mathcal{Z}})$. Then, for $k = 1, 2, \cdots, p-1$,

$$\mathcal{H}_0 : \text{rank at least } k \quad vs. \quad \mathcal{H}_1 : \text{rank at most } (k-1) \tag{9}$$

For each $k$, the hypothesis test consists of two parts. First, the noise level $\sigma_{est}(k)$ of $(\nabla_{\mathbf{z}} f)(\mathbf{z})$ should be estimated to perform a hypothesis test. The Pseudorank (Kritchman & Nadler, 2008) suggests the consistent noise estimate algorithm under the assumption that $\lambda_{k+1}, \lambda_{k+2} \cdots, \lambda_p$ are the noise

components where $\lambda_i = (\sigma_i^{\mathbf{z}})^2$. Second, we test whether $\lambda_k$ belongs to the noise components based on the corresponding Tracy-Widom distribution as follows:

$$\lambda_k \le \sigma_{est}^2(k) \left(\mu_{n,p-k} + s(\alpha) \cdot \sigma_{n,p-k}\right), \tag{10}$$

where $\alpha$ denotes a chosen confidence level. We chose $\alpha = 0.1$ in our experiments. The above test is repeated until Eq 10 is satisfied. Then, the estimated rank $K$ becomes $K = k - 1$.

**Preprocessing** The Pseudorank algorithm supposes the isotropic Gaussian noise on the Jacobian matrix. However, even considering the randomness of empirical covariance, the observed singular values of Jacobian matrix are *too sparse* ($\sigma_{min}/\sigma_{max} \approx 10^{-9}$). Hence, the isotropic Gaussian assumption leads to the underestimation of the noise level, which causes the overestimation of intrinsic rank (In our experiments, estimated rank $> 200$ and $\sigma_{rank} \approx 0.003$). To address this problem, we introduce a simple preprocessing on the singular values of the Jacobian. Before applying the Pseudorank algorithm, we filter out the singular values $\{\sigma_i^{\mathbf{z}}\}_i$ with $\{(\sigma_i^{\mathbf{z}})^2 \le \theta_{pre} \cdot \max_i(\sigma_i^{\mathbf{z}})^2\}$. We set $\theta_{pre} \in \{0.0005, 0.001, 0.005, 0.01\}$.

**Validity of the estimated local dimension** We suggest the *Off-manifold* experiment to assess the validity of the estimated local dimension. Intuitively, the intrinsic local dimension at $\mathbf{w} \in \mathcal{M}$ is the number of coordinate axes required to locally describe the learned manifold $\mathcal{M}$. Here, the tangent vector at $\mathbf{w}$ along $k$-th axis is the $k$-th LB. In this respect, **the Off-manifold experiment tests whether the latent perturbation along the $k$-th LB $\mathbf{v}_k^{\mathbf{w}}$ stays in the latent manifold $\mathcal{M} = f(\mathcal{Z})$.** If the margin of $\mathcal{M}$ at $\mathbf{w}$ in the $k$-th LB direction is large enough, then the $k$-th axis is needed to locally approximate $\mathcal{M}$. To be more specific, we solve the following optimization problem by Adam optimizer (Kingma & Ba, 2015) on MSE loss with a learning rate 0.005 for 1000 iterations for each $k$:

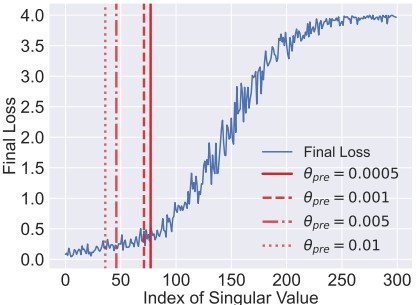

Figure 2: **Off-manifold Results** in $\mathcal{W}$-space of StyleGAN2.

$$\mathbf{w}_{init} = f(\mathbf{z}_{init}), \quad \mathbf{w}_{ptb} = \mathbf{w}_{init} + c \cdot \mathbf{v}_k^{\mathbf{w}}, \tag{11}$$

$$\mathbf{z}_{opt} = \arg\min_{\mathbf{z}} \|\mathbf{w}_{ptb} - f(\mathbf{z})\|^2 \quad \text{with} \quad \mathbf{z}_0 = \mathbf{z}_{init}. \tag{12}$$

We ran the Off-manifold experiments on $\mathcal{W}$-space of StyleGAN2. Figure 2 shows the final objective $\|\mathbf{w}_{ptb} - f(\mathbf{z}_{opt})\|^2$ after the optimization for each LB $\mathbf{v}_i^{\mathbf{w}}$ with $c = 2$. The red vertical lines denote the estimated local dimension for each $\theta_{pre}$. (See the appendix for the Off-manifold results with various $c = \|\mathbf{w}_{ptb} - \mathbf{w}_{init}\|$.) The monotonous increase in the final loss shows that $f(\mathbf{z}_{opt})$ cannot approach close to $\mathbf{w}_{ptb}$. In other words, the diameter in the $k$-th LB direction decreases as the index $k$ increases. Although there is a dependency on the preprocessing threshold, the rank estimation algorithm chooses the principal part of local manifold around $\mathbf{w}_{init}$ without overestimates as desired. Particularly, the estimated rank with $\theta_{pre} = 0.005$ appears to find a transition point of the final loss.

## 3.2 COMPARISON TO PREVIOUS RANK ESTIMATION

**Sparsity Constraint** LowRankGAN (Zhu et al., 2021) introduced a convex optimization problem called Principal Component Pursuit (PCP) (Candès et al., 2011) to find a low-rank factorization of Jacobian $(\nabla_{\mathbf{z}} f)(\mathbf{z})$ (Eq 13):

$$\text{minimize}_{L,S} \quad \|L\|_* + \gamma \cdot \|S\|_1$$
$$\text{s.t.} \quad L + S = (\nabla_{\mathbf{z}} f)^{\mathsf{T}}(\mathbf{z}) \cdot (\nabla_{\mathbf{z}} f)(\mathbf{z}). \tag{13}$$

where $\|L\|_* = \sum_i \sigma_i(L)$ is the nuclear norm, i.e. the sum of all singular values, $\|S\|_1 = \sum_{i,j} |S_{i,j}|$, and $\gamma > 0$ is a positive regularization parameter. PCP encourages the sparsity on corruption $S$ through $\ell_1$ regularizer.

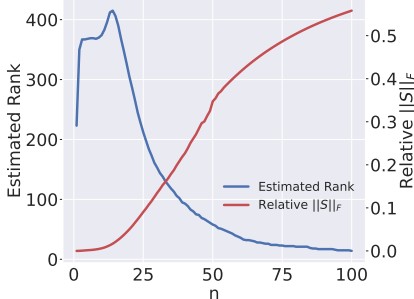

Figure 3: **Rank Estimation with Sparsity** under various $n = 1/\gamma$.

However, we believe that the sparsity assumption is not adequate for finding the intrinsic rank of Jacobian. To test the validity of the sparsity assumption, we monitored how the low-rank representation

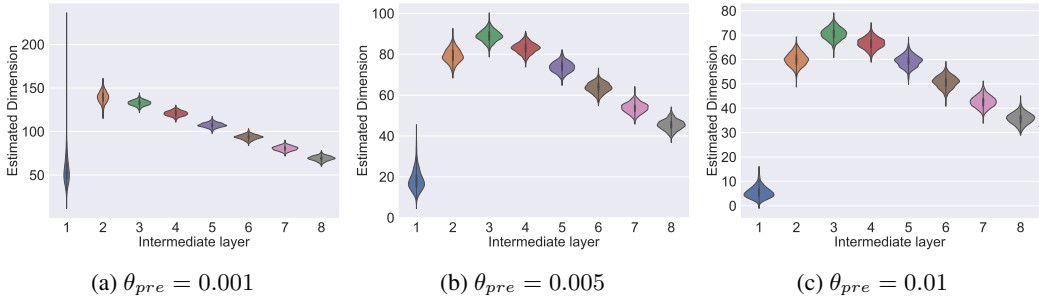

(a) $\theta_{pre} = 0.001$  (b) $\theta_{pre} = 0.005$  (c) $\theta_{pre} = 0.01$

Figure 4: **Local Dimension Distribution** of the intermediate layers in the mapping network.

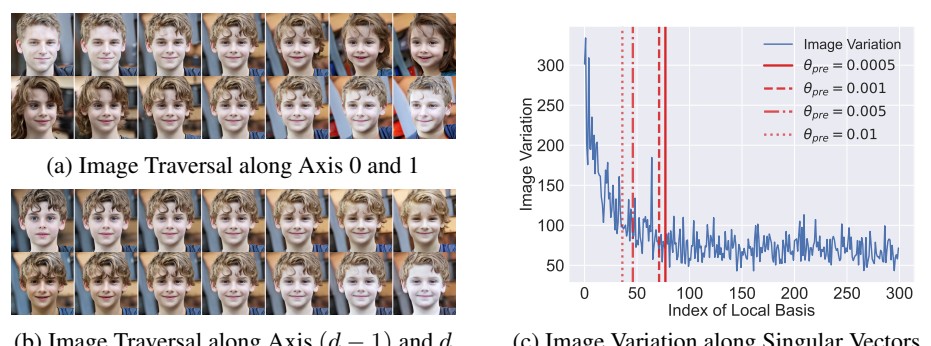

(a) Image Traversal along Axis 0 and 1

(b) Image Traversal along Axis $(d-1)$ and $d$  (c) Image Variation along Singular Vectors

Figure 5: **Local Dimension Evaluation in Image Space** where $d$ denotes the estimated local dimension with $\theta_{pre} = 0.01$. Fig 5c shows the image variation intensity $\|\nabla_{\mathbf{v}_i^w} g(\mathbf{w})\|_F$ along each LB $\mathbf{v}_i$.

$L$ changes as we vary the regularization parameter $n = 1/\gamma$ as in (Zhu et al., 2021) (Fig 3). The estimated rank decreases unceasingly without saturation as we increase $n$, i.e., refine the Jacobian stronger. We consider that the rank saturation should occur if this assumption is adequate for finding an *intrinsic* rank because it implies regularization robustness. But the low-rank factorization through PCP does not show any saturation until the Frobenius norm of corruption $\|S\|_F$ reaches over 50% of the initial matrix $\|(\nabla_{\mathbf{z}} f)^{\mathsf{T}}(\mathbf{z}) \cdot (\nabla_{\mathbf{z}} f)(\mathbf{z})\|_F$.

**Interpretation as Frobenious Norm** The Pseudorank algorithm can be interpreted as a Nuclear-Norm Penalization (NNP) problem (Eq 14) for matrix denoising (Donoho & Gavish, 2014). This NNP framework is similar to PCP in LowRankGAN except for the regularization $\|E\|_F$. While PCP requires an iterative optimization of Alternating Directions Method of Multipliers (ADMM) (Boyd et al., 2011; Lin et al., 2010), NNP provides an explicit closed-form solution $L^*$ through SVD.

$$\text{minimize}_{L,S} \quad \|L\|_* + \gamma \cdot \|E\|_F \qquad \text{s.t. } L + E = (\nabla_{\mathbf{z}} f)(\mathbf{z}). \tag{14}$$

$$\Rightarrow L^* = U \left( \Sigma - \frac{1}{2\gamma} \cdot I \right)_+ V^{\mathsf{T}} \qquad \text{where } (\nabla_{\mathbf{z}} f)(\mathbf{z}) = U\Sigma V^{\mathsf{T}} \text{ (SVD)}, \tag{15}$$

for $(M_+)_{i,j} = \max(M_{i,j}, 0)$. Therefore, the intrinsic rank estimation by NNP is determined by choosing a threshold $1/(2\gamma)$ for the singular values $\{\sigma_i^{\mathbf{z}}\}_i$ of Jacobian. The Pseudorank algorithm selects this threshold by running a series of hypothesis tests.

## 3.3 LATENT SPACE ANALYSIS OF STYLEGAN

We analyzed the intermediate layers of the mapping network in StyleGAN2 trained on FFHQ using our local dimension estimation (See Fig 11 for StyleGAN architectures). First, Figure 4 shows the distribution of estimated local dimensions for 1k samples of each intermediate layer for each $\theta_{pre}$ (See the appendix for the rank statistics under all $\theta_{pre}$). Note that the algorithm provides an unstable rank estimate on the most unsparse 1st layer (Fig 1b) under the small $\theta_{pre} \in \{0.0005, 0.001\}$. However, this phenomenon was not observed in the other layers. Hence, we focus on the layers with reasonable depth, i.e., from 3 to 8. Even though changing $\theta_{pre}$ results in an overall shift of the estimation, the

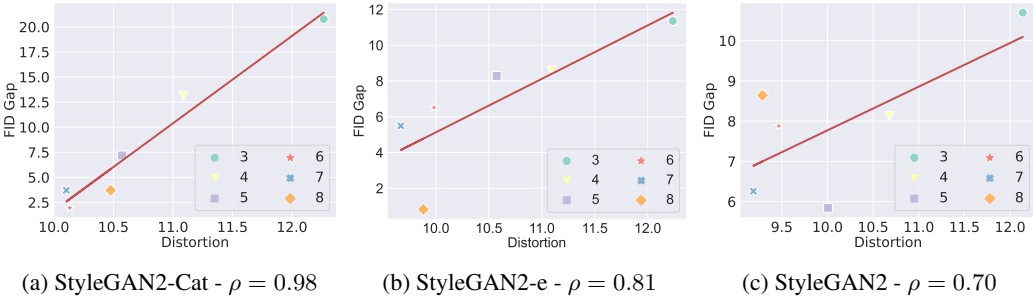

(a) StyleGAN2-Cat - $\rho = 0.98$    (b) StyleGAN2-e - $\rho = 0.81$    (c) StyleGAN2 - $\rho = 0.70$

Figure 6: **Correlation between Distortion metric (↓) and FID gap (↓)** when $\theta_{pre} = 0.005$. FID gap represents the difference between FID score of LB and the global basis (Härkönen et al., 2020). Each point represents a $i$-th intermediate layer in the mapping network.

trend and relative ordering between layers are the same. In accordance with Fig 1b, the intrinsic dimension monotonically decrease as the layer goes deeper. Second, we evaluated the estimated rank on image space (Fig 5). Figure 5a and 5b show the image traversal along the first two axes and the two axes $(d-1, d)$ around the estimated rank $d$ with $\theta_{pre} = 0.01$. Fig 5c presents the size of the directional derivative $\|\nabla_{\mathbf{v}_i^{\mathbf{w}}} g(\mathbf{w})\|_F$ along the $i$-th LB $\mathbf{v}_i^{\mathbf{w}}$ at $\mathbf{w}$, estimated by the finite difference scheme. The result shows that the estimated rank covers the major variations in the image space. One advantage of the unsupervised method over the supervised method for finding disentangled perturbation is that the discovered semantic is not restricted to the pre-defined attributes. However, we cannot know the number of discovered perturbations without additional inspections. Figure 5 shows that the estimated dimension provides an upper bound on the number of these perturbations.

## 4 UNSUPERVISED GLOBAL DISENTANGLEMENT EVALUATION

In this section, we investigate two closely related important questions on the disentanglement property of a GAN. As a reminder, *global basis* refers to the *sample-independent* semantically meaningful perturbations on a latent space, such as GANSpace and SeFa. In this regard, **the global-basis-compatibility represents how well the optimal global basis can work on the target latent space.** Specifically, the global-basis-compatibility is defined as the quality of image traversal along the optimal global basis. This is a property of the latent space itself. If the global basis does not exist in the first place, all proposed global basis can only show limited success in that latent space, no matter how we find it. Then, the two questions are as follows:

Q1. Can we evaluate the global-basis-compatibility of the latent space without posterior assessment? (Choi et al., 2022b)

Q2. Can we evaluate the disentanglement without attribute annotations? (Locatello et al., 2019)

These two questions are closely related because the ideal disentanglement includes a global basis representation where each element corresponds to the attribute-coordinate. **In this paper, the global disentanglement property of a latent space denotes this global representability along the attribute-coordinate.** To answer these questions, we propose an unsupervised global disentanglement metric, called **Distortion**. We evaluated the global-basis-compatibility by the image fidelity under global basis perturbation (Q1) and the disentanglement by semantic factorization (Q2). Our experimental results show that our proposed metric has a high correlation with the global-basis-compatibility (Q1) and the supervised disentanglement score (Q2) on various StyleGANs. (See the appendix for robustness of Distortion to $\theta_{pre}$.)

**Global Disentanglement Score**    Intuitively, our global disentanglement score assesses the inconsistency of *intrinsic* tangent space for each latent manifold. The framework of analyzing the semantic property of a latent space via its tangent space was first introduced in Choi et al. (2022b). Choi et al. (2022b) suggested this framework, inspired by the observation that each basis vector (LB) of a tangent space corresponds to a local disentangled latent perturbation. In this work, we develop this idea and propose a *layer-wise score* for global disentanglement property. Following Choi et al. (2022b), we employ the Grassmannian (Boothby, 1986) metric to measure a distance between two tangent spaces. In particular, we use a dimension-normalized version of the Geodesic Metric (Ye & Lim, 2016). We chose the Geodesic Metric instead of the Projection Metric (Karrasch, 2017) because

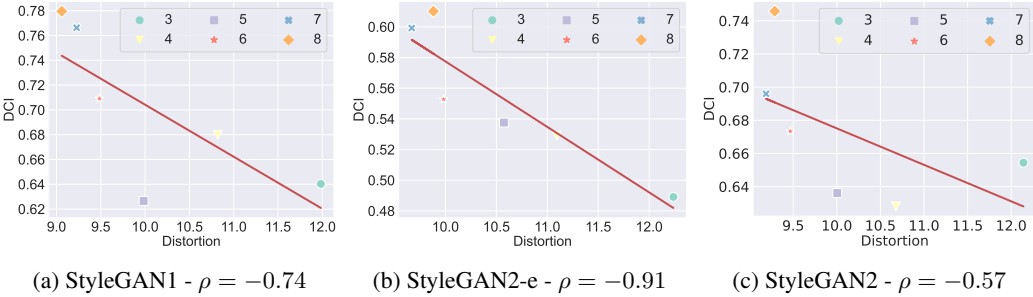

(a) StyleGAN1 - $\rho = -0.74$    (b) StyleGAN2-e - $\rho = -0.91$    (c) StyleGAN2 - $\rho = -0.57$

Figure 7: **Correlation between Distortion metric ($\downarrow$) and DCI ($\uparrow$)** when $\theta_{pre} = 0.005$. DCI (Eastwood & Williams, 2018) is a supervised disentanglement metric that requires attribute annotations.

of its better discriminability (See the appendix for detail). The dimension-normalized version is adopted because the local dimension changes according to its estimated region.

For two $k$-dimensional subspaces $W, W'$ of $\mathbb{R}^n$, let $M_W, M_{W'} \in \mathbb{R}^{n \times k}$ be the column-wise concatenation of orthonormal basis for $W, W'$, respectively. Then, the dimension-normalized Geodesic Metric is defined as $d_{\text{geo}}^k(W, W') = \left( \frac{1}{k} \sum_{i=1}^k \theta_i^2 \right)^{1/2}$ where $\theta_i = \cos^{-1}(\sigma_i(M_W^\top M_{W'}))$ denotes the $i$-th principal angle between $W$ and $W'$ for $i$-th singular value $\sigma_i$. Then, Distortion score $\mathcal{D}_\mathcal{M}$ for the latent manifold $\mathcal{M}$ is evaluated as follows:

1. To assess the overall inconsistency of $\mathcal{M}$, measure the expectation of Grassmannian distance between two intrinsic tangent spaces $T_{\mathbf{w}_i} \mathcal{M}_{\mathbf{w}_i}^{k_i}$ (Eq 5) at two **random** $\mathbf{w} \in \mathcal{M}$

$$I_{rand} = \mathbb{E}_{\mathbf{z}_i \sim p(\mathbf{z}), \mathbf{w}_i = f(\mathbf{z}_i)} \left[ d_{\text{geo}}^k \left( T_{\mathbf{w}_1} \mathcal{M}_{\mathbf{w}_1}^k, T_{\mathbf{w}_2} \mathcal{M}_{\mathbf{w}_2}^k \right) \text{ for } k = \min(k_1, k_2) \right]. \quad (16)$$

2. To normalize the overall inconsistency, measure the same Grassmannian distance between two **close** $\mathbf{w} \in \mathcal{M}$ for $\epsilon = 0.1$

$$I_{local} = \mathbb{E}_{\mathbf{z}_1 \sim p(\mathbf{z}), |\mathbf{z}_2 - \mathbf{z}_1| = \epsilon} \left[ d_{\text{geo}}^k \left( T_{\mathbf{w}_1} \mathcal{M}_{\mathbf{w}_1}^k, T_{\mathbf{w}_2} \mathcal{M}_{\mathbf{w}_2}^k \right) \text{ for } k = \min(k_1, k_2) \right]. \quad (17)$$

3. ***Distortion*** of $\mathcal{M}$ is defined as the relative inconsistency $\mathcal{D}_\mathcal{M} = I_{rand}/I_{local}$.

**Distortion and Global Disentanglement**    In this paragraph, we clarify *why the globally disentangled latent space shows a low Distortion score.* Assume a latent space $\mathcal{M}$ is globally disentangled. Then, there exists an optimal global basis of $\mathcal{M}$, where each basis vector corresponds to an image attribute on the entire $\mathcal{M}$. By definition, this optimal global basis is the local basis at all latent variables. Assuming that LB finds the local basis Choi et al. (2022b), each global basis vector would correspond to one LB vector at each latent variable. In this regard, our local dimension estimation finds a principal subset of LB, which includes these corresponding basis vectors. In conclusion, if the latent space is globally disentangled, this principal set of LB at each latent variable would contain the common global basis vectors. Hence, the intrinsic tangent spaces would contain the common subspace generated by this common basis, which leads to a small Grassmannian metric between them. Therefore, the global disentanglement of the latent space leads to a low Distortion score.

**Global-Basis-Compatibility**    We tested whether Distortion $\mathcal{D}_\mathcal{M}$ is meaningful in estimating the global-basis-compatibility. We chose GANSpace as a reference global basis because of its broad applicability. The global basis proposed in GANSpace is PCA components of latent variable samples (Härkönen et al., 2020). Hence, we can find a global basis in any intermediate layers. We chose FID (Heusel et al., 2017) gap between LB and GANSpace as a measure of global-basis-compatibility. FID is measured for 50k samples of perturbed images along the 1st component of LB and GANSpace, respectively. Distortion metric is tested on StyleGAN2 on LSUN Cat (Yu et al., 2015), StyleGAN2 with configs E and F (Karras et al., 2020b) on FFHQ to test the generalizability of correlation to the global-basis-compatibility. StyleGAN2 in Fig 6 denotes StyleGAN2 with config F because config F is the usual StyleGAN2 model. The perturbation intensity is set to 5 in LSUN Cat and 3 in FFHQ. Distortion metric shows a strong positive correlation of 0.98, 0.81 and 0.70 to FID gap in Fig 6. This result demonstrates that Distortion metric can be an unsupervised criterion for selecting the latent space with high global-basis-compatibility. Before finding a global basis, we can use Distortion metric as a prior investigation for selecting an appropriate target latent space.

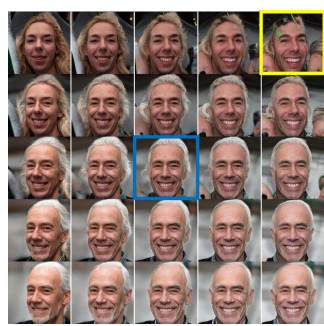 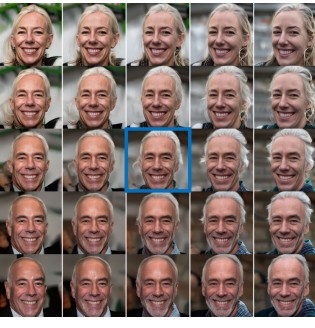 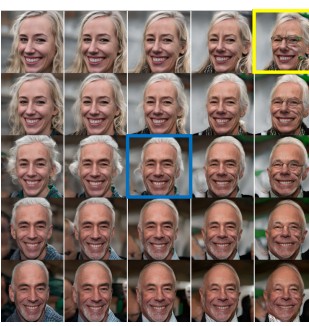

(a) Max-distorted layer 3      (b) Min-distorted layer 7      (c) Layer 8 ($\mathcal{W}$-space)

Figure 8: **Subspace Traversal on the intermediate layers** along the global basis. The upper-right corner of max-distorted layer 3 and layer 8 show visual artifacts. However, the min-distorted layer 7 does not show such a failure. The initial image (center) is traversed along the 1st (horizontal) and 2nd (vertical) components of GANSpace.

**Disentanglement Score** We assessed a correlation between the unsupervised Distortion metric and a supervised disentanglement score. Following the work of Wu et al. (2020), we adopted DCI score (Eastwood & Williams, 2018) as the supervised disentanglement score for evaluation, and employed 40 binary attribute classifiers pre-trained on CelebA (Liu et al., 2015) to label generated images. Each DCI score is assessed on 10k samples of latent variables with the corresponding attribute labels. In Fig 7, StyleGAN1, StyleGAN2-e, and StyleGAN2 refer to StyleGAN1 and StyleGAN2s with config E and F trained on FFHQ. Note that DCI experiments are all performed on FFHQ because the DCI score requires attribute annotations. DCI and Distortion metrics show a strong negative correlation on StyleGAN1 and StyleGAN2-e. The correlation is relatively moderate on StyleGAN2. This moderate correlation is because Distortion metric is based on the Grassmannian metric. The Grassmannian metric measures the distance between tangent spaces, while DCI is based on their specific basis. Even if the tangent space is identical so that Distortion becomes zero, DCI can have a relatively low value depending on the choice of basis. Hence, in StyleGAN2, the high-distorted layers showed low DCI scores, but the low-distorted layers showed relatively high variance in DCI score. Nevertheless, the strong correlation observed in the other two experiments suggests that, in practice, the basis vector corresponding to a specific attribute has a limited variance in a given latent space. Therefore, Distortion metric can be an unsupervised indicator for the supervised disentanglement score.

**Traversal Comparison** For a visual comparison of the global-basis-compatibility, we observed the image traversal along the global basis on the max-distorted layer 3, min-distorted layer 7, and layer 8 ($\mathcal{W}$-space) of StyleGAN2 on FFHQ. Our global-basis-compatibility result implies that the global basis would perform better in terms of image fidelity on the min-distorted layer. To impose a more challenging condition, we introduced the subspace traversal (Choi et al., 2022b) along the first and second components of the global basis with a perturbation intensity 4. In Fig 8, the global basis shows visual artifacts at the corners in the subspace traversal on the max-distorted layer 3 and $\mathcal{W}$-space. Nevertheless, the min-distorted layer 7 shows the stable traversal without any failure. This result proves that comparing Distortion scores can be a criterion for selecting a better latent space with higher global-basis-compatibility. (See the appendix for additional results.)

## 5 CONCLUSION

In this paper, we proposed a local intrinsic dimension estimation algorithm for the intermediate latent space in a pre-trained GAN. Using this algorithm, we analyzed the intermediate layers in the mapping network of StyleGANs on various datasets. Moreover, we suggested an unsupervised global disentanglement metric called Distortion. The analysis of the mapping network demonstrates that Distortion metric shows a high correlation between the global-basis-compatibility and disentanglement score. Although finding an optimal preprocessing hyperparameter $\theta_{pre}$ was beyond the scope of this work, the proposed metric showed robustness to the hyperparameter. Moreover, our local dimension estimation scheme has the potential to be applied to various models. For example, the adversarial robustness of the classifier can be analyzed by projecting the adversarial noise onto the estimated feature space. We consider this kind of research would be an interesting future research.

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

# A    DEFINITION OF DISENTANGLED LATENT SPACE

**Disentangled perturbation**    In the GAN disentanglement literature, several studies investigated the disentanglement property of the latent space by finding disentangled perturbations that make a disentangled transformation of an image in one generative factor, such as GANSpace (Härkönen et al., 2020), SeFa (Shen & Zhou, 2021), and Local Basis (Choi et al., 2022b). To be more specific, for a latent variable $z \in \mathcal{Z} \subset \mathbb{R}^d$, let $f = (f_1, f_2, \cdots, f_d)$ be a generative factor of $G(z)$ where $G$ denotes the generator. $T_j(x)$ denotes a transformation of an image $x$ in the $j$-th generative factor. The disentangled perturbation $v_j(z)$ for the base latent variable $z$ on the $j$-th generative factor is defined as follows (The perturbation intensity $\|v_j(z)\|$ and the corresponding change in $j$-th generative factor $\triangle f_j$ is omitted for brevity.):

$$G(z + v_j(z)) = T_j(G(z)). \tag{18}$$

In this paper, the global basis refers to the sample-independent disentangled perturbations on a latent space:

$$v_j(z) = v_j \quad \text{for all } z \in Z. \tag{19}$$

For example, consider a pre-trained GAN model that generates face images. Then, the disentangled perturbation in this model is the latent perturbation direction that make the generated face change only in the wrinkles or hair color as presented in Härkönen et al. (2020). This disentangled perturbation is the global basis if all generated images show the same semantic variation when latent perturbed along it.

**Disentangled space**    The (globally) disentangled latent space is defined in terms of disentangled perturbations. The latent space is globally disentangled if there exists the global basis for the generative factors of data. In other words, for each generative factor $f_j$ for $1 \leq j \leq d$, there exists a corresponding latent perturbation direction $v_j$ such that all latent variables show the semantic variation in $f_j$ when perturbed along $v_j$. Then, we can interpret the vector component of this global basis $v_j$ as having a correspondence with the $j$-th generative factor $f_j$.

$$f_j \longleftrightarrow c_j \quad \text{when } z = \sum_{1 \leq j \leq d} c_j \cdot v_j \text{ and } f_j \text{ denotes the } j\text{-th generative factor of } G(z). \tag{20}$$

In this paper, we described the above correspondence as the representation of globally disentangled latent space in the attribute-coordinate in Sec 4. This is consistent with the definition of disentanglement, introduced in Bengio et al. (2013). For example, consider the dSprites dataset. The dSprites is a synthetic dataset consisting of two-dimensional shape images, which is widely used for disentanglement evaluation. The generative factors of dSprites are shape, scale, orientation, position on the x-axis, and position on the y-axis. Then, the (globally) disentangled latent space for the dSprites dataset is a five-dimensional vector space $Z = \mathbb{R}^5$ where $c_1$ represents the shape, $c_2$ represents the scale, and so on.

# B    NOISE ESTIMATION OF PSEUDORANK ALGORITHM

For completeness, we include the convergence theorem for the largest eigenvalue of the empirical covariance matrix for Gaussian noise in Johnstone (2001) and the noise estimation algorithm provided in Kritchman & Nadler (2008).

**Theorem 1** ((Johnstone, 2001)). *The distribution of the largest eigenvalue $\lambda_1$ of the empirical covariance matrix for $n$-samples of $\mathcal{N}(0, I_p)$ converges to a Tracy-Widom distribution:*

$$P\left(\lambda_1 < \sigma^2 \left(\mu_{n,p} + s \cdot \sigma_{n,p}\right)\right) \to F_\beta(s) \quad \text{as } n, p \to \infty \text{ with } c = p/n \text{ fixed.} \tag{21}$$

$$\text{where} \quad \mu_{n,p} = \frac{1}{n} \left(\sqrt{n - \frac{1}{2}} + \sqrt{p - \frac{1}{2}}\right)^2, \tag{22}$$

$$\sigma_{n,p} = \frac{1}{n} \left(\sqrt{n - \frac{1}{2}} + \sqrt{p - \frac{1}{2}}\right) \left(\frac{1}{\sqrt{n - 1/2}} + \frac{1}{\sqrt{p - 1/2)}}\right)^{1/3}, \tag{23}$$

*where $F_\beta$ denotes the Tracy-Widom distribution of order $\beta = 1$ for real-valued observations.*

**Algorithm** Solve the following non-linear system of $K+1$ equations involving the $K+1$ unknowns $\hat{\rho}^1, \cdots, \hat{\rho}^K$ and $\sigma_{est}^2$:

$$\sigma_{\text{KN}}^2 - \frac{1}{p-K}\left[\sum_{j=K+1}^{p}\lambda_j + \sum_{j=1}^{K}(\lambda_j - \hat{\rho}_j)\right] = 0, \tag{24}$$

$$\hat{\rho}_j^2 - \hat{\rho}_j\left(\lambda_j + \sigma_{est}^2 - \sigma_{est}^2\frac{p-K}{n}\right) + \lambda_j\sigma_{est}^2 = 0. \tag{25}$$

This system of equations can be solved iteratively. Check Kritchman & Nadler (2008) for detail.

## C   RELATION BETWEEN RANK ESTIMATION ALGORITHM AND OPTIMIZATION

**Theorem 2.** *The following optimization problem, called Nuclear-Norm Penalization (NNP),*

$$minimize_{L,S} \quad \|L\|_* + \gamma \cdot \|E\|_F \qquad s.t. \ L + E = (\nabla_{\mathbf{z}} f)(\mathbf{z}), \tag{26}$$

*has a solution*

$$L^* = U\left(\Sigma - \frac{1}{2\gamma}\cdot I\right)_+ V^\mathsf{T}, \tag{27}$$

*where $(\nabla_{\mathbf{z}} f)(\mathbf{z}) = U\Sigma V^\mathsf{T}$ (SVD) and $(M_+)_{i,j} = \max(M_{i,j}, 0)$.*

*Proof.* Denote $Y := \nabla_{\mathbf{z}} f$ and $h(L) := \|L\|_* + \gamma\|Y - L\|_F$. We want to show that $L_*$ minimizes $h(L)$. Then, the necessary and sufficient condition for this is:

$$0 \in \partial h(\widehat{L_*}) = \{2\gamma(L_* - Y) + z : z \in \partial\|\widehat{L_*}\|_*\} \iff 2\gamma(Y - L_*) \in \partial\|\widehat{L_*}\|_*. \tag{28}$$

Note that $Y = U\Sigma V^\mathsf{T}$ and $\hat{L} = U(\Sigma - \frac{1}{2\gamma}I)_+ V^\mathsf{T}$. We can write

$$Y = U_1\Sigma_1 V_1^\mathsf{T} + U_2\Sigma_2 V_2^\mathsf{T}, \tag{29}$$

with $\text{diag}(\Sigma_1) > \frac{1}{2\gamma}$ and $\text{diag}(\Sigma_2) \leq \frac{1}{2\gamma}$. Then,

$$\widehat{L_*} = U_1\left(\Sigma - \frac{1}{2\gamma}I\right)V_1^\mathsf{T}, \tag{30}$$

$$Y - \widehat{L_*} = U_2\Sigma_2 V_2^\mathsf{T} + \frac{1}{2\gamma}U_1 V_1^\mathsf{T} = \frac{1}{2\gamma}(U_1 V_1^\mathsf{T} + 2\gamma U_2\Sigma_2 V_2^\mathsf{T}). \tag{31}$$

By doing tedious calculation, we can verify that $U_1 V_1^\mathsf{T} + 2\gamma U_2\Sigma_2 V_2^\mathsf{T}$ meet the condition of Lemma 1, so that $U_1 V_1^\mathsf{T} + 2\gamma U_2\Sigma_2 V_2^\mathsf{T} \in \partial\|\widehat{L_*}\|$. Therefore, $0 \in \partial h(\widehat{L_*})$ and it completes the proof. $\square$

**Lemma 1.** *Let $X \in \mathbb{R}^{m\times n}$ and $f(x) = \|X\|_*$. Then,*

$$\partial f(X) = \partial\|X\|_* = \{Z \in \mathbb{R}^{m\times n} : \|Z\|_2 \leq 1 \ and \ \langle Z, X\rangle = \|X\|_*\}. \tag{32}$$

*Proof.* If $Z \in \partial f(X)$, then

$$f(Y) \geq f(X) + \langle Z, Y - X\rangle, \qquad \forall Y \in \mathbb{R}^{m\times n}, \tag{33}$$

$$\Leftrightarrow \langle Z, X\rangle - \|X\|_* \geq \langle Z, Y\rangle - \|Y\|_*, \quad \forall Y \in \mathbb{R}^{m\times n}, \tag{34}$$

$$\Leftrightarrow \langle Z, X\rangle - \|X\|_* \geq \sup_{Y\in\mathbb{R}^{m\times n}}(\langle Z, Y\rangle - \|Y\|_*) = \begin{cases} 0, & \text{if } \|Z\|_2 \leq 1, \\ \infty, & \text{otherwise.} \end{cases} \tag{35}$$

And $0 \leq \langle Z, X, \rangle - \|X\|_* = \langle Z, X, \rangle - \sup_{\|M\|_2\leq 1}\langle M, X, \rangle \leq 0$, thus $\langle Z, X, \rangle = \|X\|_*$. $\square$

# D  GRASSMANNIAN METRIC FOR DISTORTION - GEODESIC VS. PROJECTION

Our proposed Distortion metric $\mathcal{D} = I_{rand}/I_{local}$ is defined as the relative inconsistency of intrinsic tangent spaces on a latent manifold (Sec 4). The inconsistency ($I_{rand}, I_{local}$) is measured by the Grassmannain (Boothby, 1986) distance between tangent spaces, particularly by Geodesic Metric (Ye & Lim, 2016). In this section, we present why we choose the Geodesic Metric instead of the Projection Metric (Karrasch, 2017) among Grassmannian distances. Informally, the Geodesic Metric provides a better discriminability compared to the Projection Metric. For completeness, we begin with the definitions of the Grassmannian manifold and two distances defined on it.

**Definitions**  Let $V$ be the $n$-dimensional vector space. The *Grassmannian manifold* $\mathrm{Gr}(k, V)$ (Boothby, 1986) is defined as the set of all $k$-dimensional linear subspaces of $V$. Then, for two $k$-dimensional subspaces $W, W' \in \mathrm{Gr}(k, V)$, two Grassmannian metrics are defined as follows:

$$d_{\mathrm{proj}}(W, W') = \|P_W - P_{W'}\|, \qquad d_{\mathrm{geo}}(W, W') = \left(\sum_{i=1}^{k} \theta_i^2\right)^{1/2}. \tag{36}$$

For the *Projection Metric* $d_{\mathrm{proj}}(W, W')$, $P_W$ and $P_{W'}$ denote the projection into each subspaces and $\|\cdot\|$ represents the operator norm. For the *Geodesic Metric* $d_{\mathrm{geo}}(W, W')$, $\theta_i$ denotes the $i$-th principal angle between $W$ and $W'$. To be more specific, $\theta_i = \cos^{-1}(\sigma_i(M_W^\top M_{W'}))$ where $M_W, M_{W'} \in \mathbb{R}^{n \times k}$ are the column-wise concatenation of orthonormal basis for $W, W'$ and $\sigma_i$ represents the $i$-th singular value.

**Experiments**  To test the discriminability of these two metrics, we designed a simple experiment. Let $W, W'$ be the two 50-dimensional subspaces of $\mathbb{R}^{512}$ because the dimension of intermediate layers in the mapping network is 512. We measure the Grassmannian distance between two subspaces as we vary $\dim(W \cap W') = k_0$,

$$W = \langle e_1, e_2, \cdots, e_k \rangle, \qquad W = \langle \{e_1, e_2, \cdots, e_{k_0}\} \cup \{e_{k+1}, \cdots, e_{2k-k_0}\} \rangle \tag{37}$$

where $\{e_i\}_{1 \le i \le n}$ denotes the standard basis of $\mathbb{R}^n$. Fig 9 reports the results. The Geodesic Metric reflects the degree of intersection between two subspaces. As we increase the dimension of intersection, the Geodesic Metric decreases. However, the Projection Metric cannot discriminate the intersected dimension until it reaches the entire space.

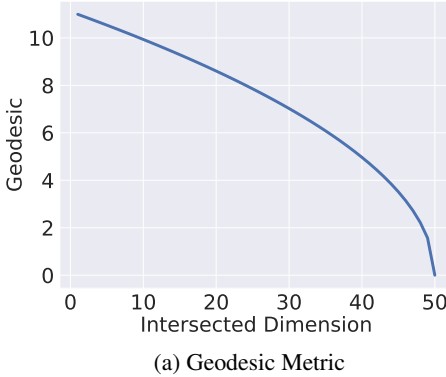

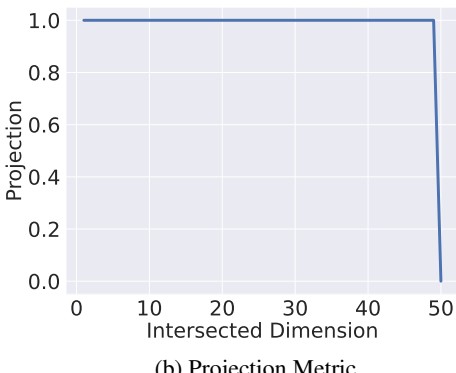

(a) Geodesic Metric  (b) Projection Metric

Figure 9: **Grassmannian metric** between two 50-dimensional subspaces $W, W' \in \mathrm{Gr}(50, \mathbb{R}^{512})$ for each intersected dimension $k_0 = \dim(W \cap W')$. While the Geodesic Metric monotonically decreases as more dimensions intersect, the Projection Metric cannot discriminate $0 \le k_0 \le 49$.

# E    ROBUSTNESS TO PREPROCESSING

In this section, we assessed the robustness of Distortion $\mathcal{D}_\mathcal{M}$ to preprocessing hyperparameter $\theta_{pre}$. Figure 10 presents the distribution of 1k samples of distortion before taking an expectation, i.e., $\left(d^k_{\mathrm{geo}}\left(T_{\mathbf{w}_1}\mathcal{M}^k_{\mathbf{w}_1}, T_{\mathbf{w}_2}\mathcal{M}^k_{\mathbf{w}_2}\right)/I_{local}\right)$, for each intermediate layer. In Fig 10, increasing $\theta_{pre}$ makes an overall translation of Distortion. However, the relative ordering between the layers remains the same. The low Distortion score of layer 8 provides an explanation for the superior disentanglement of $\mathcal{W}$-space observed in many literatures (Karras et al., 2019; Härkönen et al., 2020). Moreover, the results suggest that the min-distorted layer 7 can serve as a similar-or-better alternative.

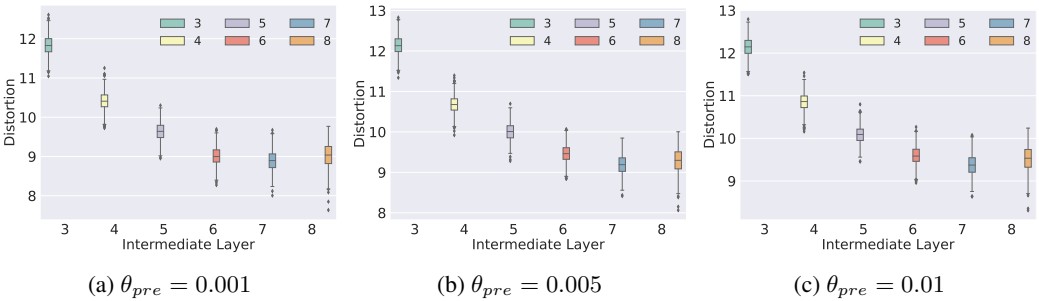

(a) $\theta_{pre} = 0.001$              (b) $\theta_{pre} = 0.005$              (c) $\theta_{pre} = 0.01$

Figure 10: **Robustness of Distortion metric $\mathcal{D}$ ($\downarrow$) to** $\theta_{pre}$ of StyleGAN2 on FFHQ.

# F    ARCHITECTURE DIAGRAM OF STYLEGANS

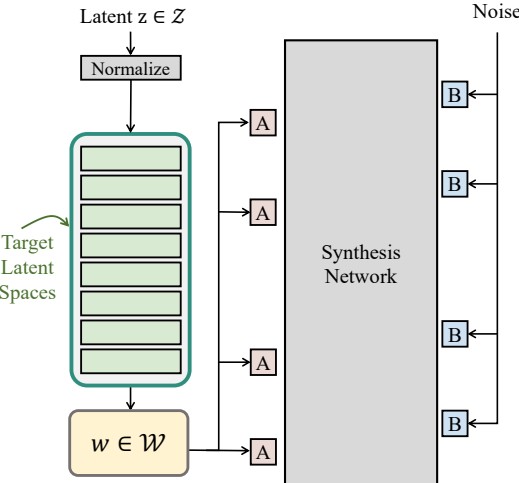

Figure 11: **Architecture of StyleGANs.** Our analysis in Sec 4 is performed in the intermediate layers of the mapping network.

# G    ADDITIONAL EXPERIMENTAL RESULTS

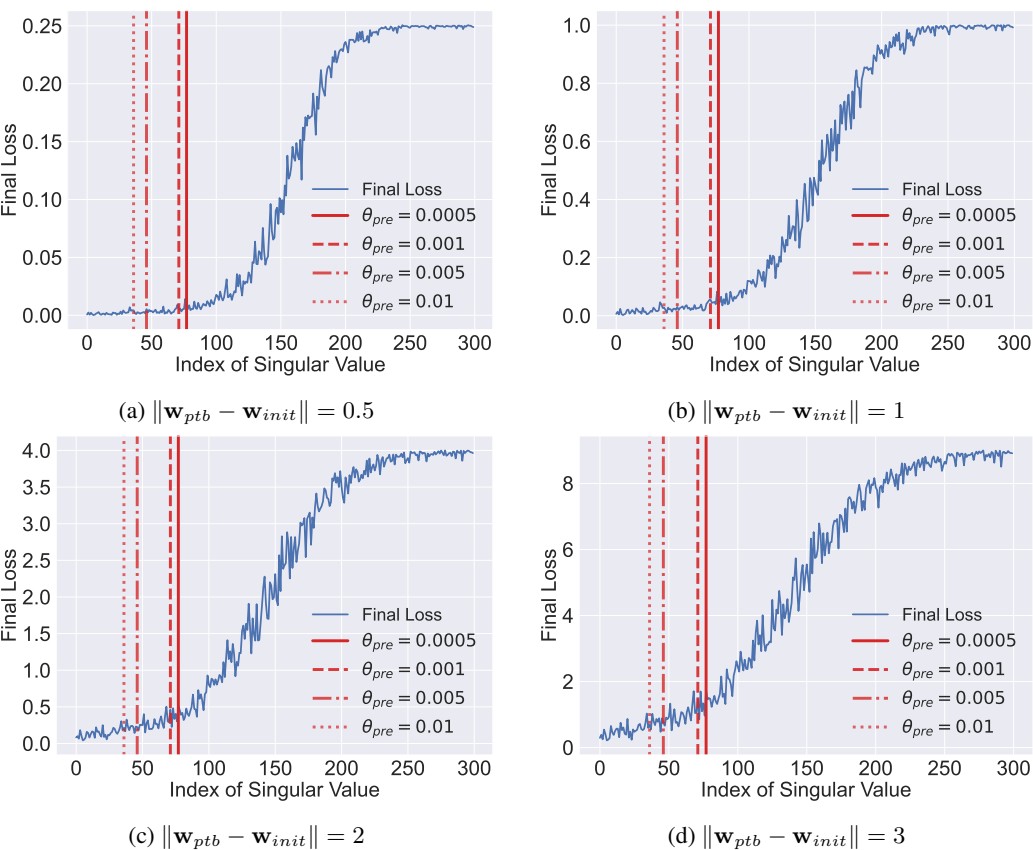

Figure 12: **Off-manifold Results** in $\mathcal{W}$-space of StyleGAN2 on FFHQ. The red vertical lines denote the estimated local dimension for various $\theta_{pre}$. The small final loss implies that the linear perturbation along that Local Basis component stays inside the learned latent manifold. For every perturbation intensity, there is a transition point from a slow increase to a sharp increase, which is interpreted as an escape from the manifold. The results demonstrate that the proposed dimension estimation algorithm finds a reasonable point without crossing the transition point. In our case, these results are interpreted as choosing the principal part of the manifold without overestimating its dimension.

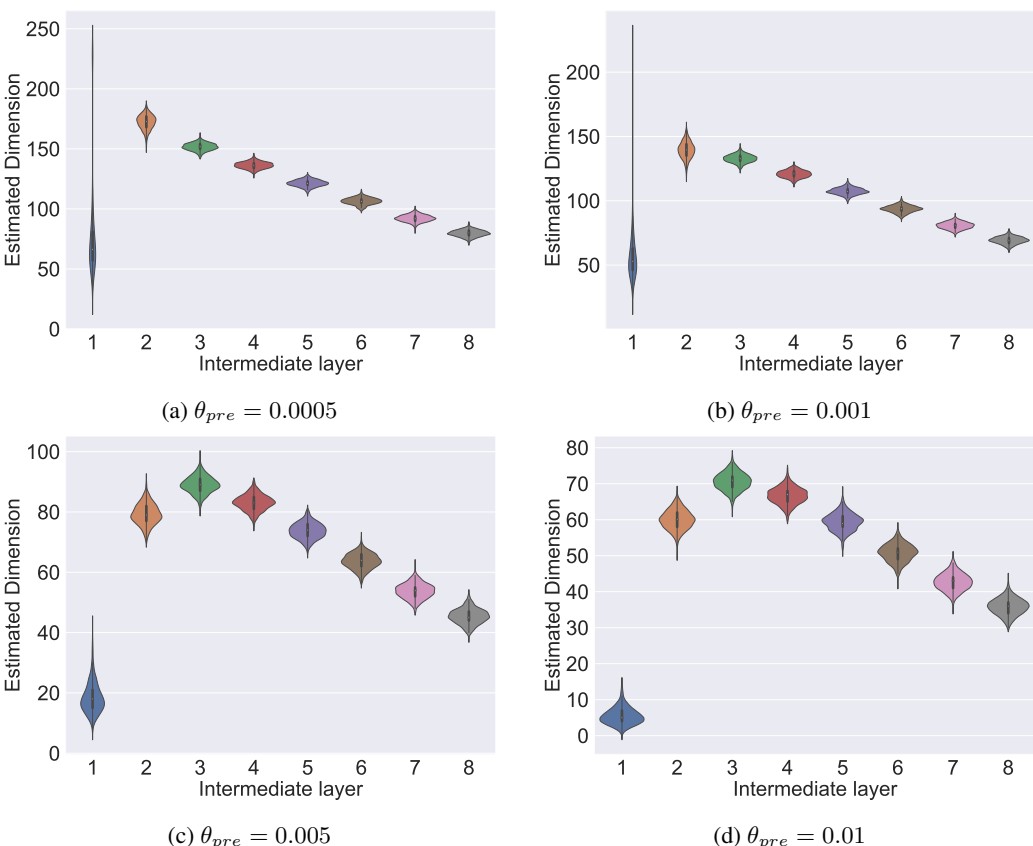

Figure 13: **Local Dimension Distribution** of the intermediate layers in the mapping network of StyleGAN2 on FFHQ. Each figure presents the distribution of estimated local dimension at each intermediate layer as we vary $\theta_{pre}$. The distributions are illustrated for 1k samples, respectively. The algorithm gives a rather unstable dimension estimate on the most unsparse first layer (Fig 1b) due to its isotropic gaussian assumption. However, this phenomenon is not observed in the layers with moderate depth, i.e., from 3 to 8. As we introduce the higher preprocessing ratio $\theta_{pre}$, the algorithm gives more strict, i.e., smaller, dimension estimates. Nevertheless, the relative trend between layers is the same. The deeper the latent manifold, the smaller its dimension.

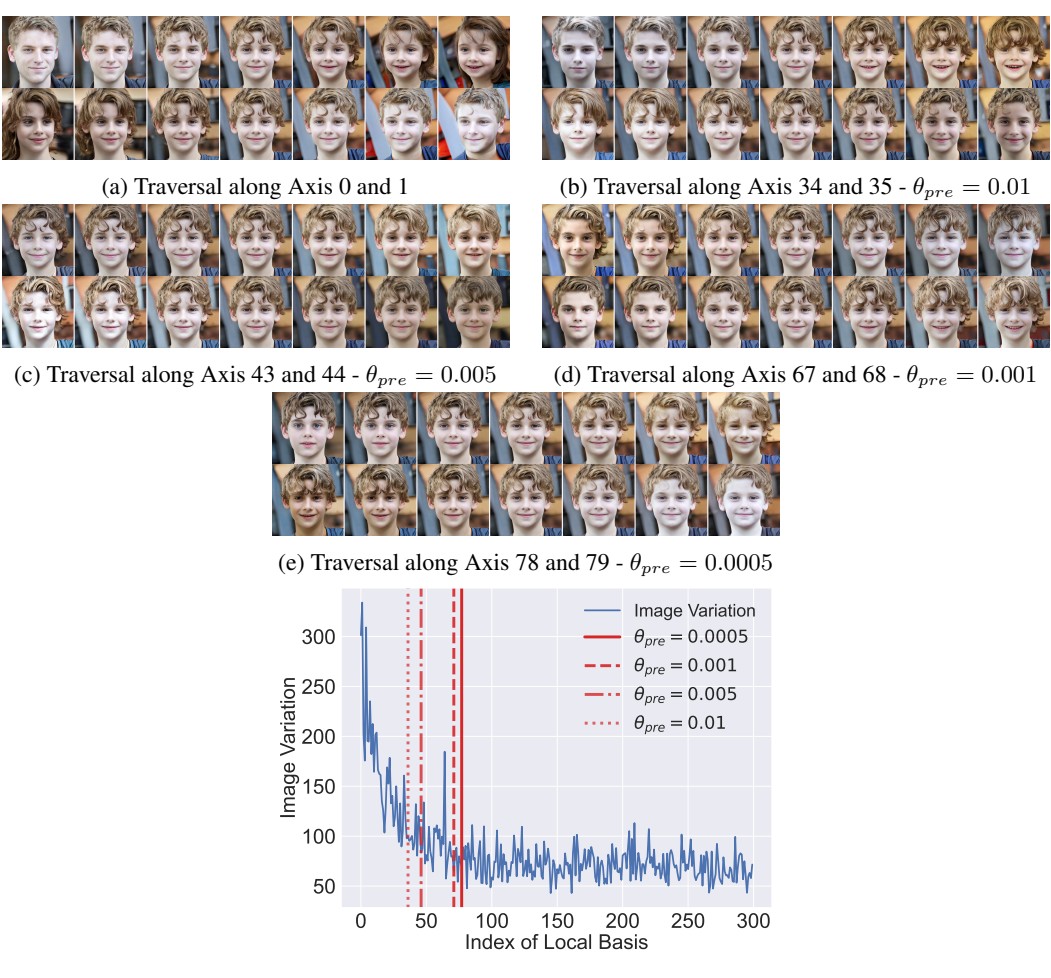

(a) Traversal along Axis 0 and 1

(b) Traversal along Axis 34 and 35 - $\theta_{pre} = 0.01$

(c) Traversal along Axis 43 and 44 - $\theta_{pre} = 0.005$

(d) Traversal along Axis 67 and 68 - $\theta_{pre} = 0.001$

(e) Traversal along Axis 78 and 79 - $\theta_{pre} = 0.0005$

(f) Image variation intensity

Figure 14: **Local Dimension Evaluation in Image Space** of StyleGAN2 on FFHQ. Figure 14b-14e show image traversals along the $(d-1)$-th and $d$-th axis where $d$ denotes the estimated local dimension for each $\theta_{pre}$. Fig 14f presents the image variation intensity $\|\nabla_{\mathbf{v}_i^{\mathbf{w}}} g(\mathbf{w})\|_F$ along each Local Basis $\mathbf{v}_i$. $\|\nabla_{\mathbf{v}_i^{\mathbf{w}}} g(\mathbf{w})\|_F$ is evaluated by the finite difference with stepsize=0.01. Note that while Local Basis is discovered by analyzing only the subnetwork from the input to target latent space, the corresponding image variation monotonically decreases and saturates. Moreover, the estimated local dimension includes the major variations in the image space. This can be observed in the image traversals. The image traversals around the estimated dimension (Fig 14b-14e) show much smaller image variations compared to Axis 0 and 1 (Fig 14a).

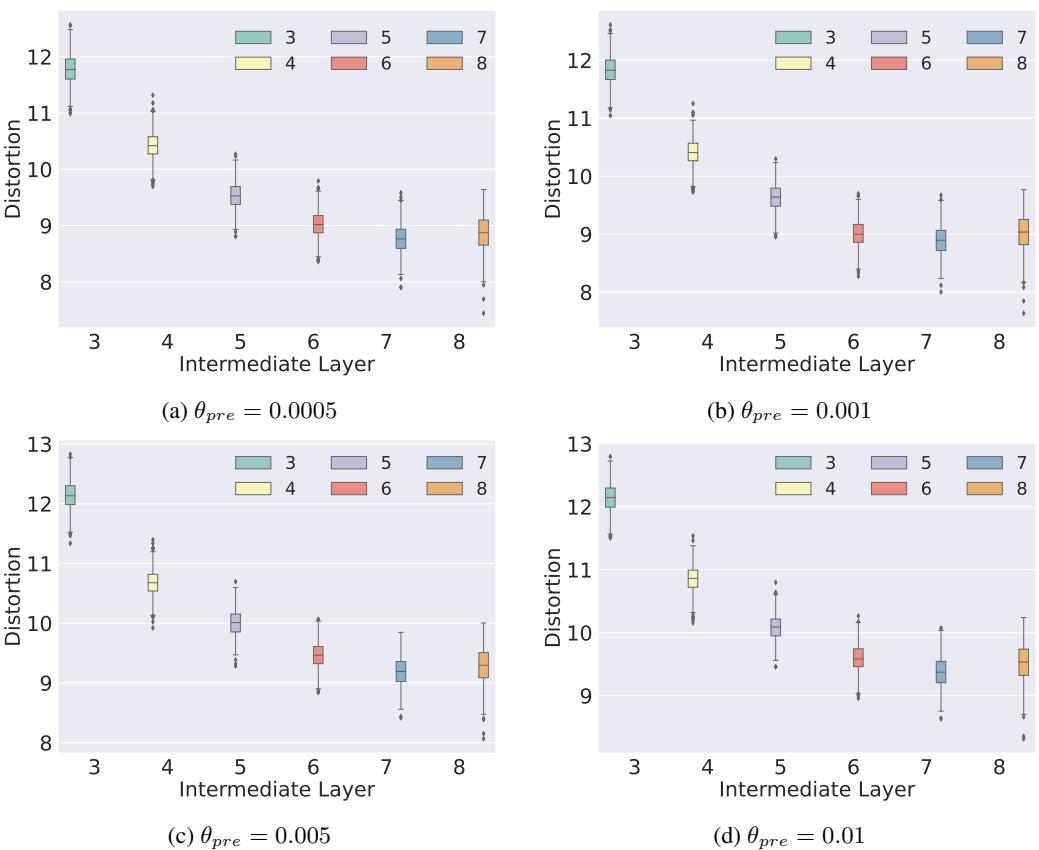

Figure 15: **Robustness of Distortion metric** $\mathcal{D}$ **to** $\theta_{pre}$ of StyleGAN2 on FFHQ. Each boxplot shows the distribution of 1k samples of Distortion before taking an average, i.e., $\left(d_{\text{geo}}^k\left(T_{\mathbf{w}_1}\mathcal{M}_{\mathbf{w}_1}^k, T_{\mathbf{w}_2}\mathcal{M}_{\mathbf{w}_2}^k\right)/I_{local}\right)$ (Sec 4), for each intermediate layer. Increasing $\theta_{pre}$ makes a slight increase in Distortion for all layers. Nevertheless, the relative ordering between layers is robust under the change of $\theta_{pre}$.

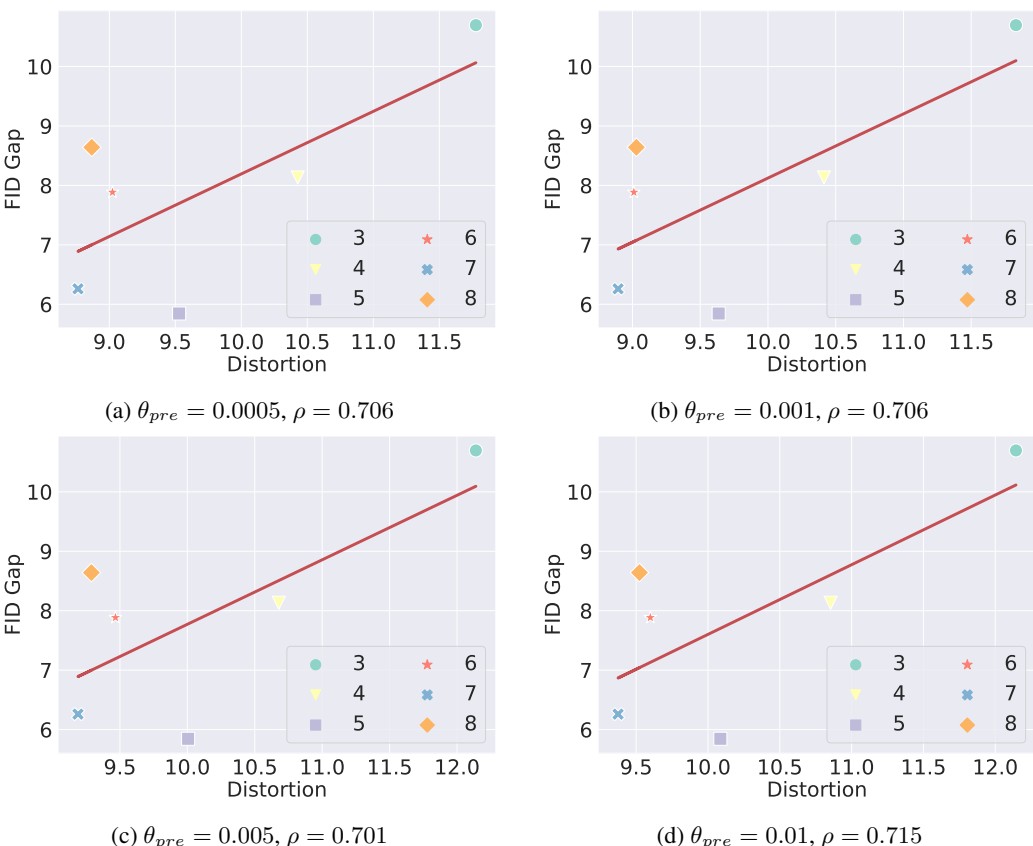

Figure 16: **Correlation between Distortion metric and FID gap** of StyleGAN2 on FFHQ. FID gap represents the difference between FID (Heusel et al., 2017) score of Local Basis (Choi et al., 2022b) and the global basis (Härkönen et al., 2020). Each FID score is measure for 50k samples of latent-perturbed images along the first component of Local Basis and GANSpace. The perturbation intensity is fixed to 3. Each point represents a $i$-th intermediate layer in the mapping network, and the red-line illustrates the linear regression of these points. $\rho$ denotes the Pearson correlation coefficient of Distortion and FID. The positive correlation between Distortion and FID remains robust regardless of $\theta_{pre}$.

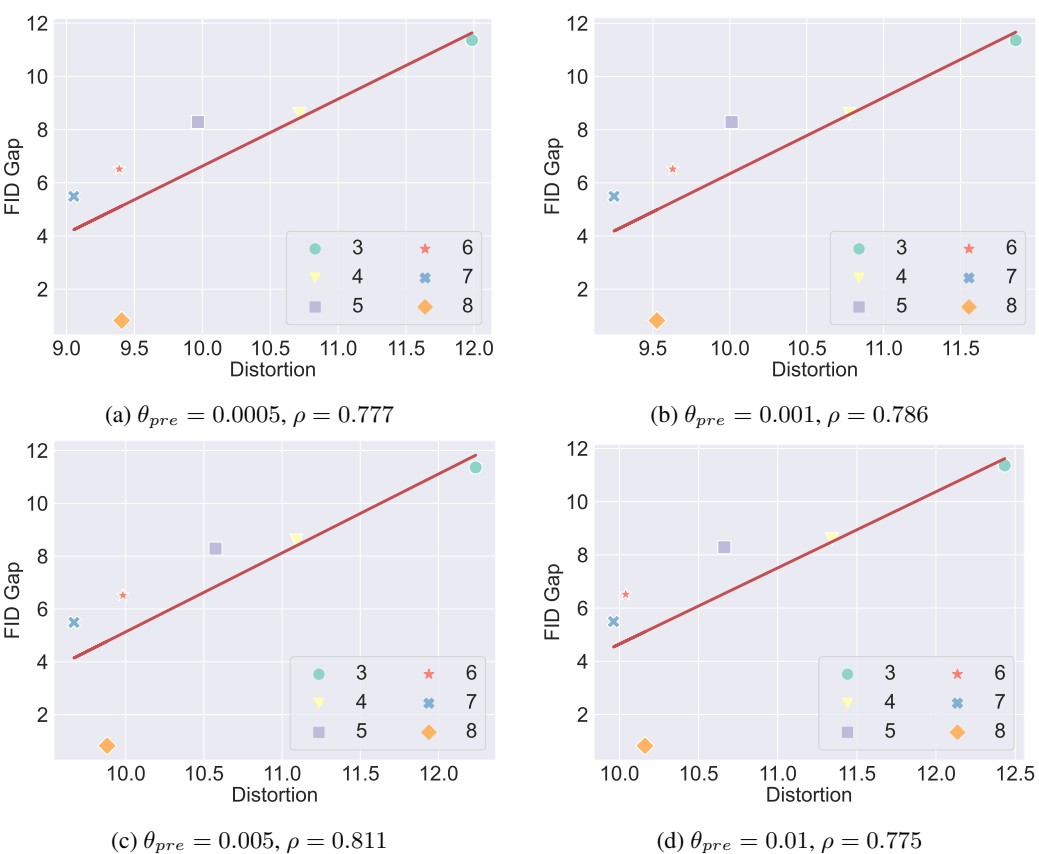

(a) $\theta_{pre} = 0.0005$, $\rho = 0.777$

(b) $\theta_{pre} = 0.001$, $\rho = 0.786$

(c) $\theta_{pre} = 0.005$, $\rho = 0.811$

(d) $\theta_{pre} = 0.01$, $\rho = 0.775$

Figure 17: **Correlation between Distortion metric and FID gap** of StyleGAN2 with config E on FFHQ.

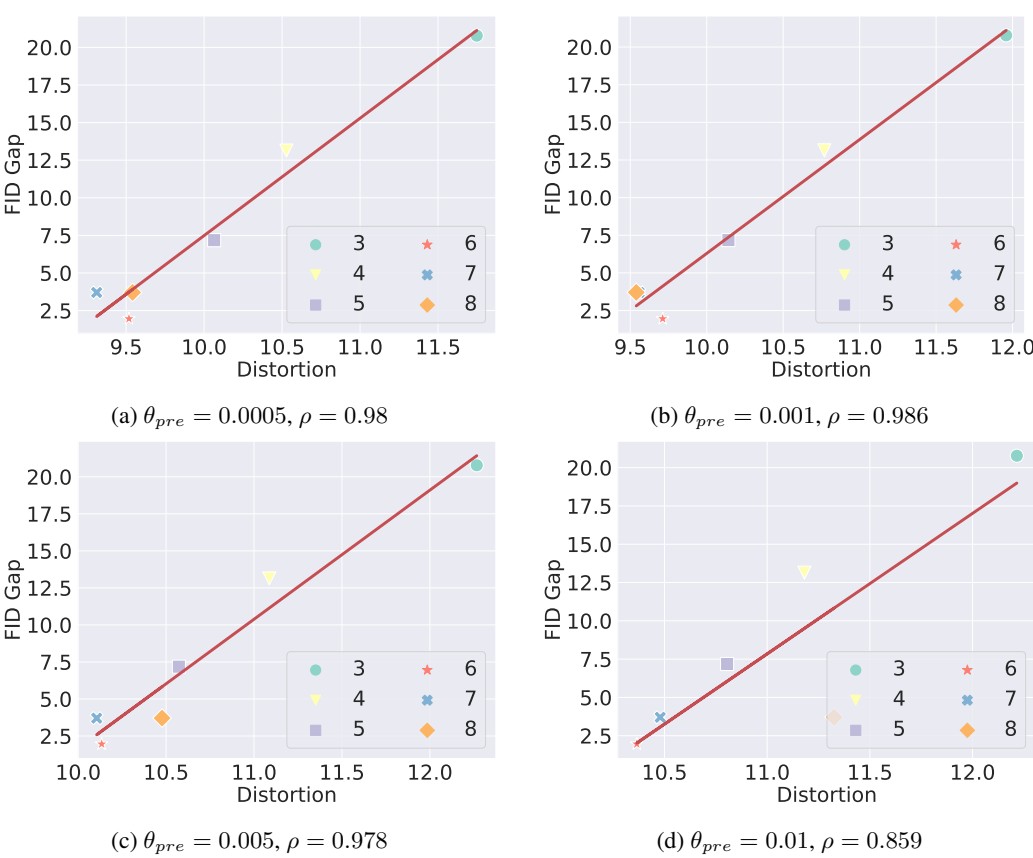

(a) $\theta_{pre} = 0.0005, \rho = 0.98$

(b) $\theta_{pre} = 0.001, \rho = 0.986$

(c) $\theta_{pre} = 0.005, \rho = 0.978$

(d) $\theta_{pre} = 0.01, \rho = 0.859$

Figure 18: **Correlation between Distortion metric and FID gap** of StyleGAN2 on LSUN Cat.

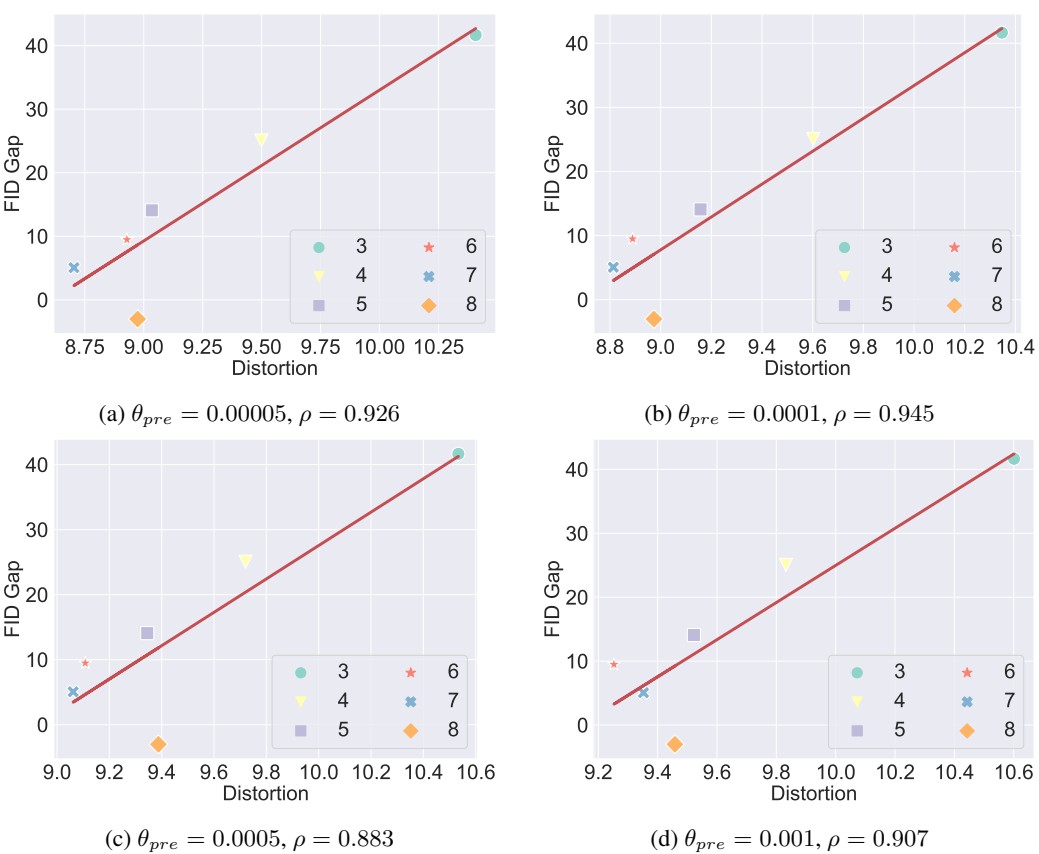

Figure 19: **Correlation between Distortion metric and FID gap** of StyleGAN2 on LSUN Horse.

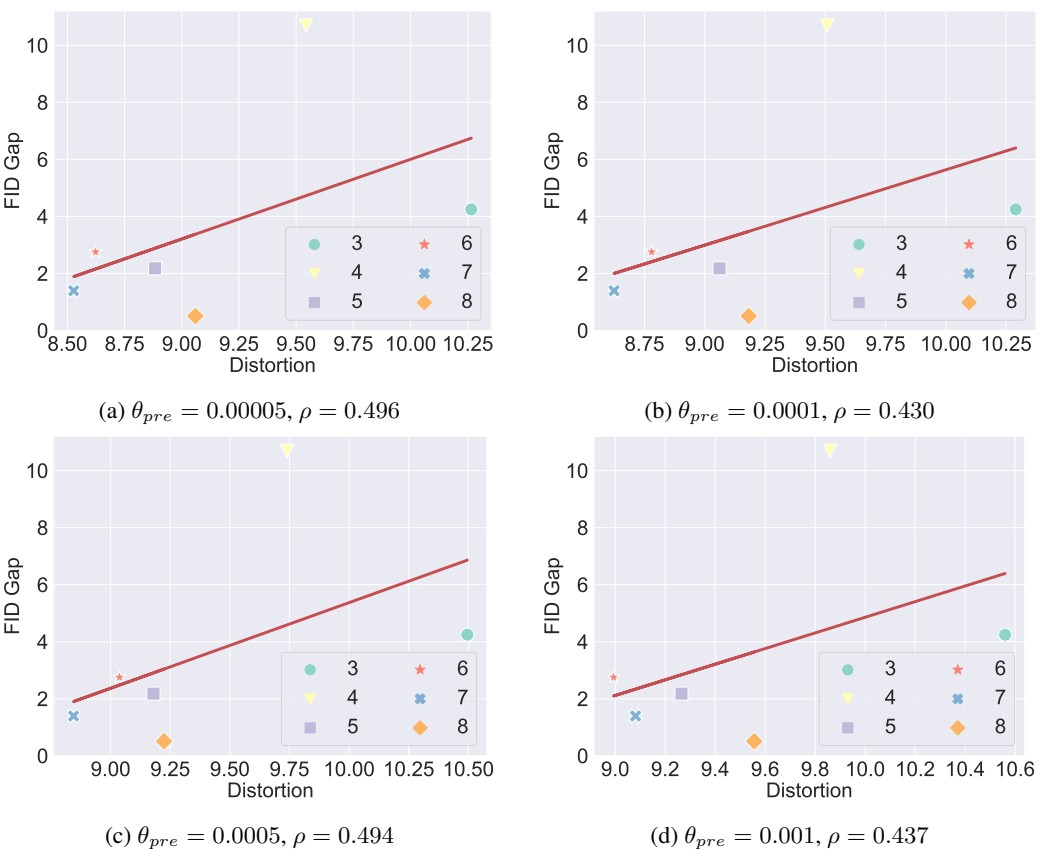

Figure 20: **Correlation between Distortion metric and FID gap** of StyleGAN2 on LSUN Church.

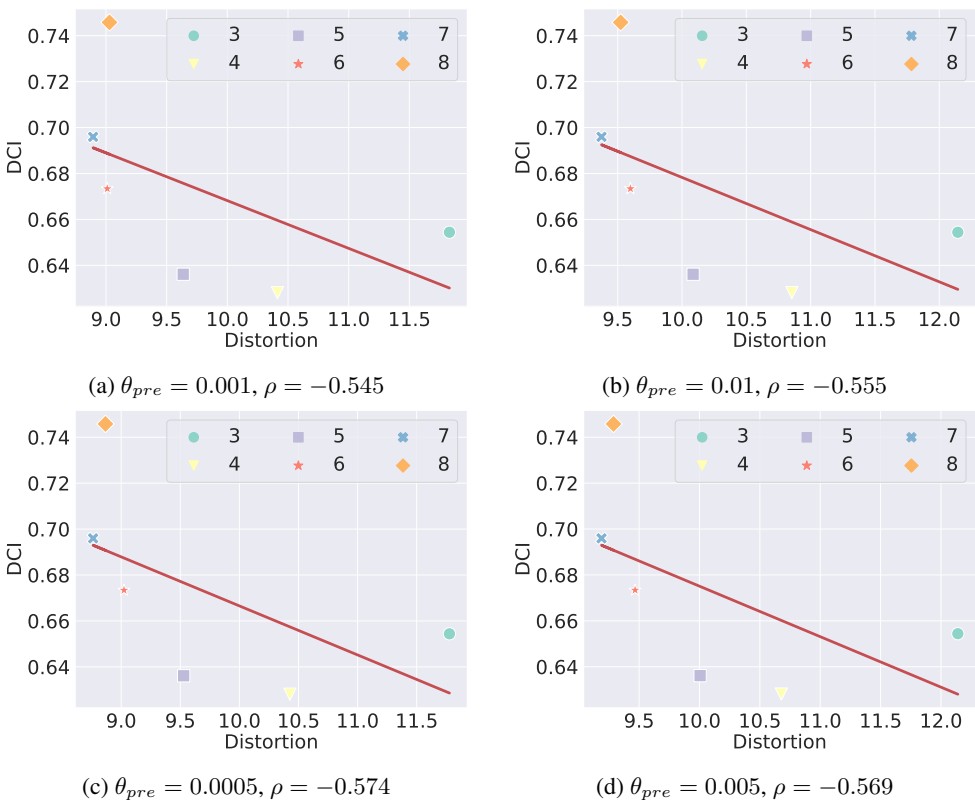

Figure 21: **Correlation between Distortion metric and DCI** of StyleGAN2 on FFHQ. Each DCI (Eastwood & Williams, 2018) score is evaluated for 10k samples of generated images, while the attribute label is generated by 40 attribute classifiers pre-trained on CelebA (Liu et al., 2015). As in Fig 16, each point and red-line represents the intermediate layers and linear regression, respectively. The Distortion and DCI score show a negative correlation.

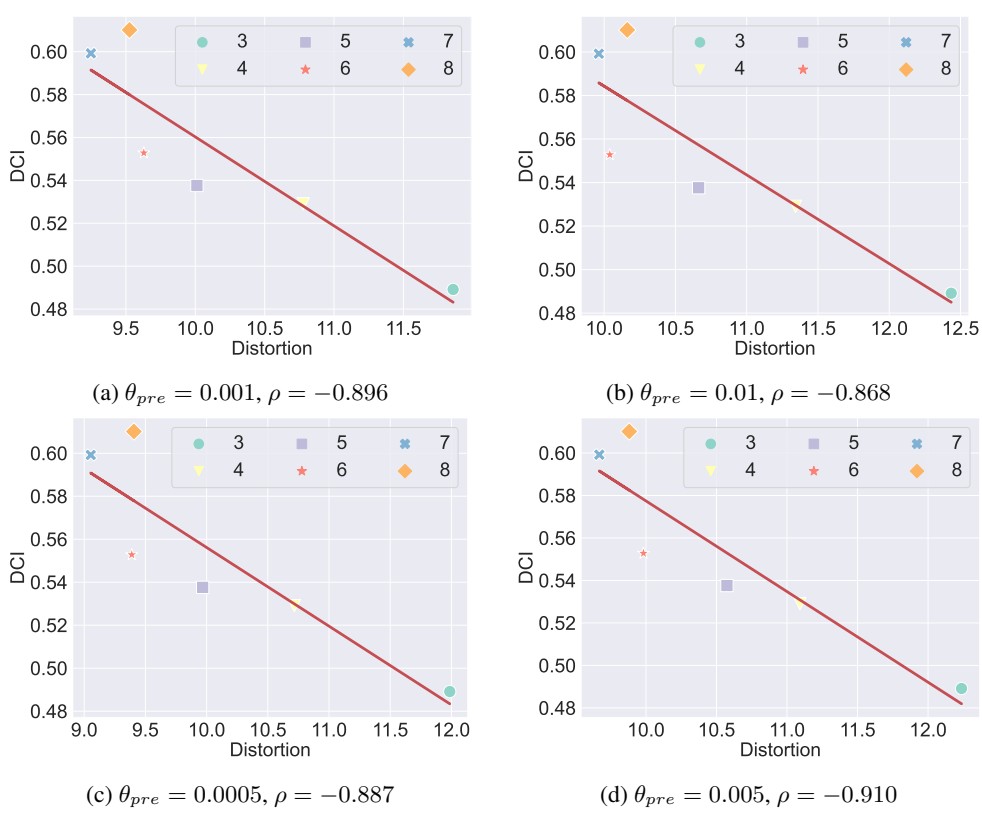

Figure 22: **Correlation between Distortion metric and DCI** of StyleGAN2 with config E on FFHQ.

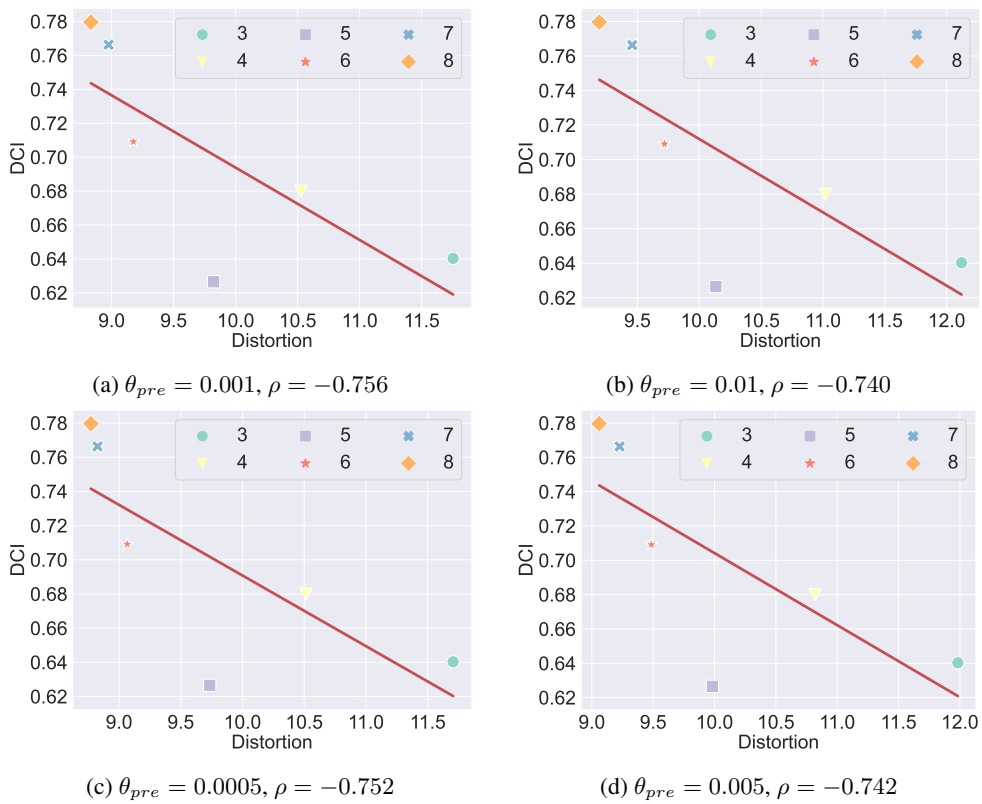

Figure 23: **Correlation between Distortion metric and DCI** of StyleGAN1 on FFHQ.

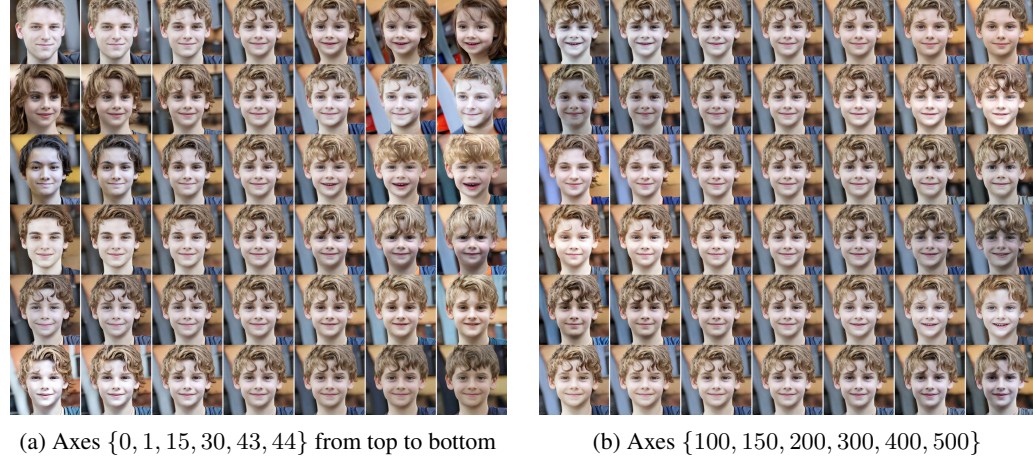

(a) Axes $\{0, 1, 15, 30, 43, 44\}$ from top to bottom  (b) Axes $\{100, 150, 200, 300, 400, 500\}$

Figure 24: **Linear Traversals along the various LB axis** on StyleGAN2-FFHQ. At this latent variable, the estimated dimension with $\theta_{pre} = 0.01$ is $d = 44$. When we traverse along the axis $i \gg d$, the images present minor variations compared to the axis $i \le d$. Traversals along the 1 44 axes display various semantic variations such as gender, skin color, age, etc.

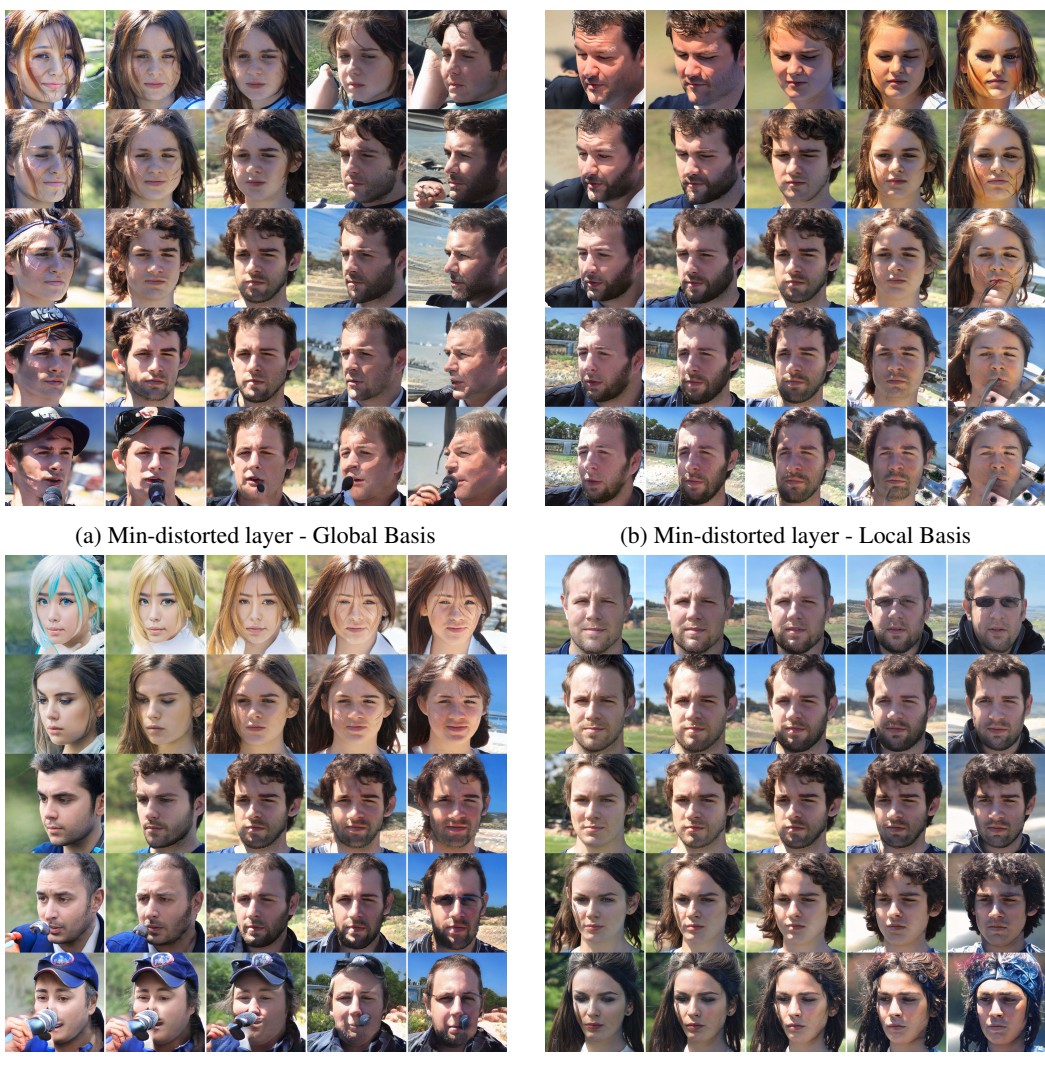

(a) Min-distorted layer - Global Basis          (b) Min-distorted layer - Local Basis

(c) Max-distorted layer - Global Basis          (d) Max-distorted layer - Local Basis

Figure 25: **Subspace Traversal (Choi et al., 2022b) on the min-distorted (7th) and max-distorted (3rd) intermediate layers** along the global basis (Härkönen et al., 2020) and Local Basis (Choi et al., 2022b) of StyleGAN2 on FFHQ. The initial image (center) is traversed along the 1st (horizontal) and 2nd (vertical) components of the chosen traversal directions with the perturbation intensity 9. The global basis shows a decent image quality on the min-distorted layer, similar to Local Basis. However, on the max-distorted layer, the subspace traversal along global basis exhibits significant failures at corners, such as image collapse (lower-left), visual artifacts (lower-right), and unnatural transformations (top-left).

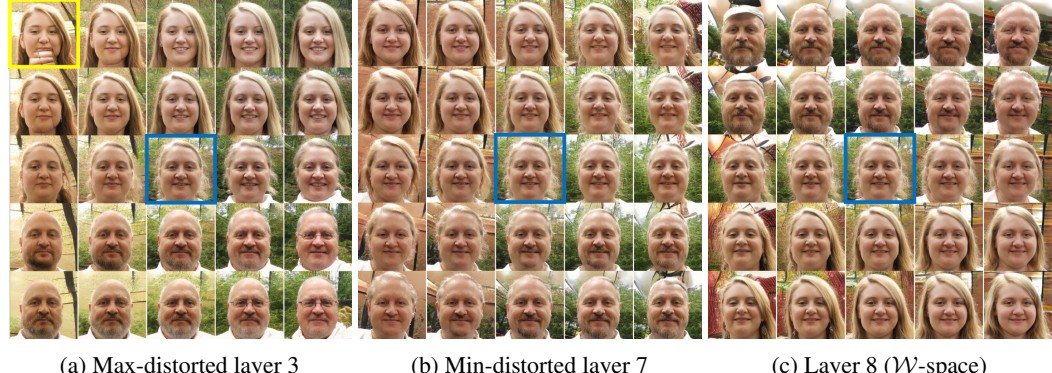

(a) Max-distorted layer 3        (b) Min-distorted layer 7        (c) Layer 8 ($\mathcal{W}$-space)

Figure 26: **Subspace Traversal** on StyleGAN2-FFHQ. The upper-left corner of layer 3 is severely deteriorated.

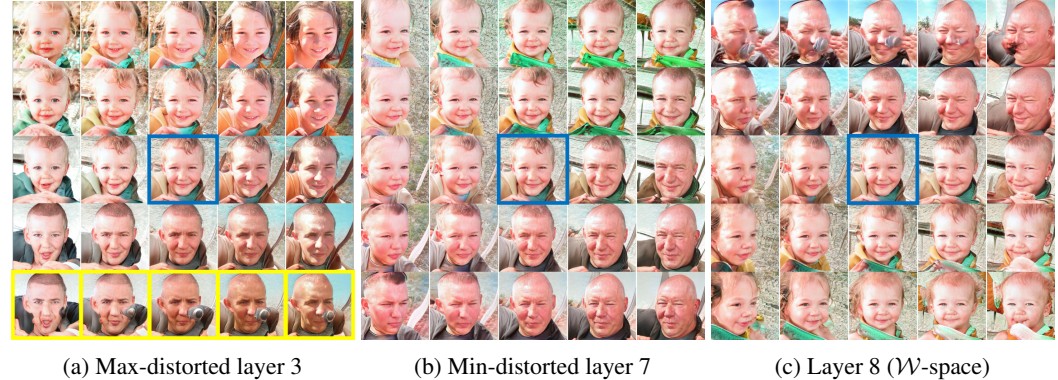

(a) Max-distorted layer 3        (b) Min-distorted layer 7        (c) Layer 8 ($\mathcal{W}$-space)

Figure 27: **Subspace Traversal** on StyleGAN2-FFHQ. The lower sides of layer 3 are severely deteriorated. Also, the upper sides of layer 8 are more deteriorated than layer 7.

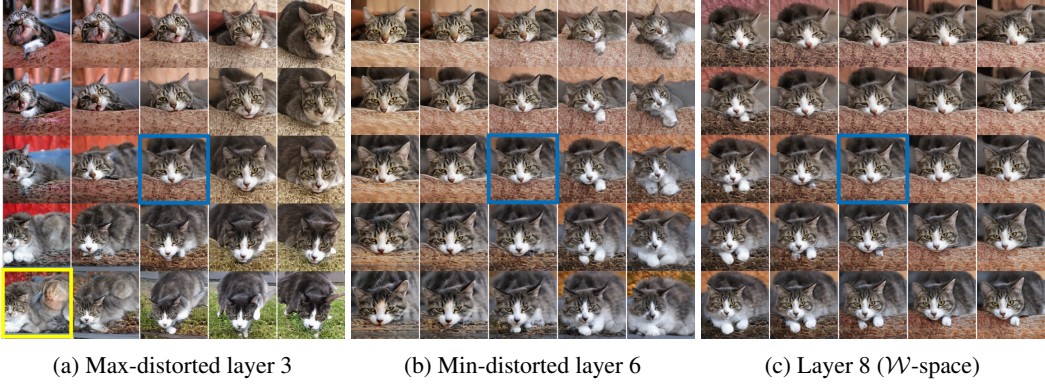

(a) Max-distorted layer 3        (b) Min-distorted layer 6        (c) Layer 8 ($\mathcal{W}$-space)

Figure 28: **Subspace Traversal** on StyleGAN2-LSUN Cat. The lower-left corner of layer 3 are severely deteriorated.

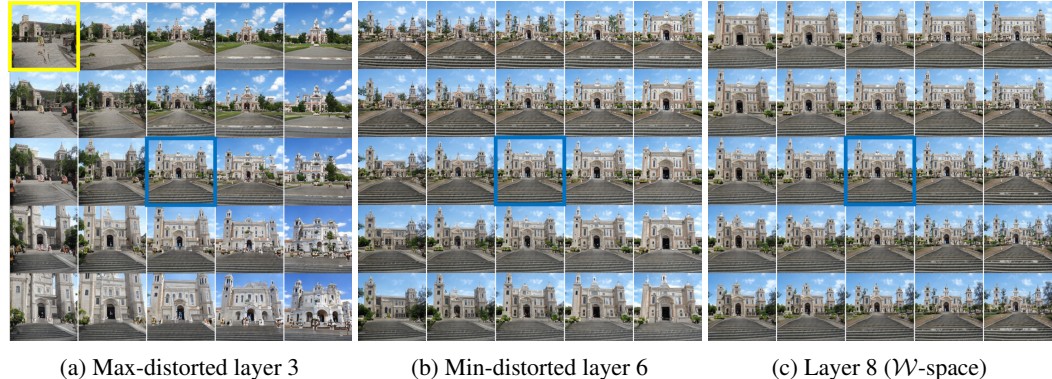

(a) Max-distorted layer 3  (b) Min-distorted layer 6  (c) Layer 8 ($\mathcal{W}$-space)

Figure 29: **Subspace Traversal** on StyleGAN2-LSUN Church. The upper-left corner of layer 3 are severely deteriorated.

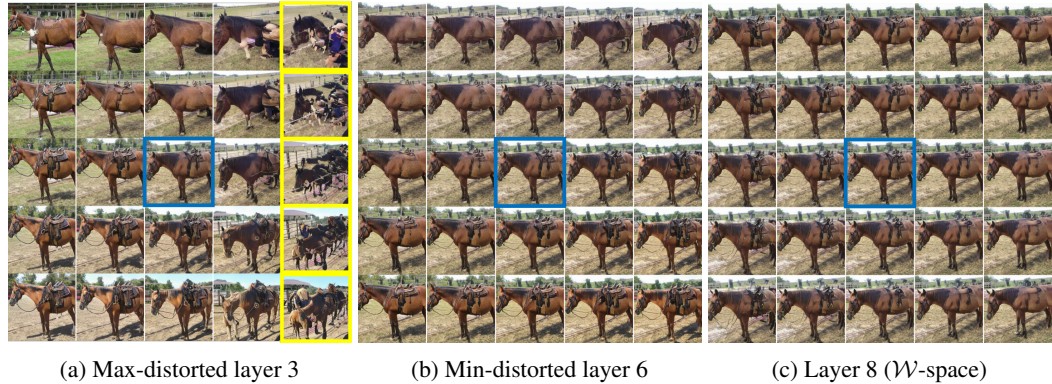

(a) Max-distorted layer 3  (b) Min-distorted layer 6  (c) Layer 8 ($\mathcal{W}$-space)

Figure 30: **Subspace Traversal** on StyleGAN2-LSUN Horse. The right sides of layer 3 are severely deteriorated.

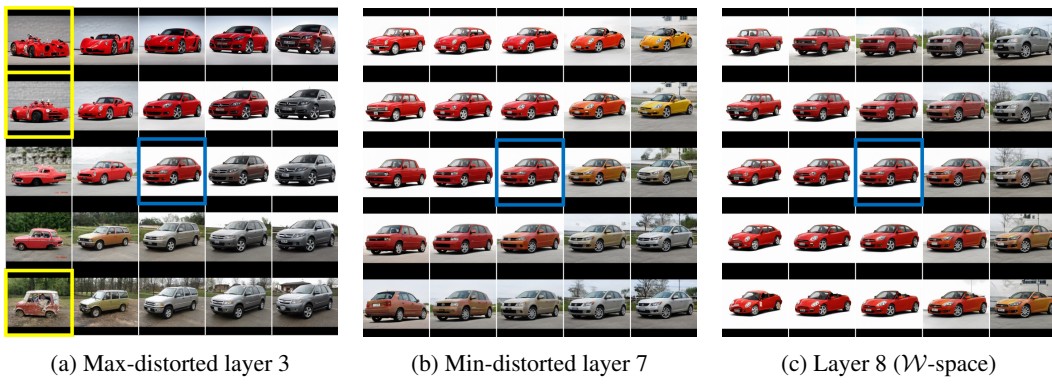

(a) Max-distorted layer 3  (b) Min-distorted layer 7  (c) Layer 8 ($\mathcal{W}$-space)

Figure 31: **Linear Traversal** on StyleGAN2-LSUN Cars. The left sides of layer 3 are severely deteriorated.

