# OpenReview forum: "Analyzing the Latent Space of GAN through Local Dimension Estimation"
_ICLR.cc/2023/Conference — Submitted to ICLR 2023_

### Official Review · Reviewer_1W36 · 2022-10-23

**Confidence:** 2
**Correctness:** 2
**Technical Novelty And Significance:** 3
**Empirical Novelty And Significance:** Not applicable
**Recommendation:** 5

**Clarity, Quality, Novelty And Reproducibility:**

- The work seems original and a non-trivial extension of Choi, 2022.
- The clarity of presentation needs much improvement.
- The lack of clarity makes it hard to fully judge the substance of the paper.
- The methods appear to be reproducible, but code was not provided.

**Strength And Weaknesses:**

Strengths:
- The paper deals with an important problem of illuminating the high-dimensional W space of StyleGAN.
- The proposed local intrinsic dimensionality estimation scheme appears original and could prove useful in latent space analysis.
- Experiments support the claims (with disclaimers below) and support the hypothesis that the proposed metric is correlated with the other two scores (global basis compatibility and supervised disentanglement score)

Weaknesses:
- The relevance of the main claims is not clearly explained. The significance of the proposed metric, in comparison to e.g. (Choi, 2022) appears to be that it is unsupervised, but if this is really the key benefit, I could not find this point clearly explained or compared to (Choi, 2022).
- The paper as a whole is difficult to follow. The writing is part of the problem, with phrases such as "The compatibility with the global basis represents how well the (optimal) global basis can work on the target latent space", "an unsupervised selection of globally disentangled
latent space among the intermediate latent spaces in a GAN", or "the number of semantically disentangled local perturbation from w".
- For the same reasons, it is not clear whether the experiments are sufficient to justify the main claims. For instance, for visual comparison, there are a few traversed layers shown in Fig. 9, it's not clear how much ground does this kind of visualization intend to cover.
- In the empirical section, the explanations are hard to follow, with sentences like "The global basis proposed in GANSpace is PCA components of latent variable samples", or "Because this optimal global basis is also the local disentangled perturbations at all latent variables".


**Summary Of The Paper:**

The paper proposes a new metric for GAN latent space estimation, called Distortion. The metric builds on the Local Basis method of (Choi, 2022), and is shown to be correlated with the "global-basis-compatibility" and supervised disentanglement score from (Eastwood & Williams, 2018). It provides an estimate of "intrinsic dimensionality" of each layer in a network.

**Summary Of The Review:**

In terms of technical contents, the paper appears potentially valuable as a contribution to GAN latent space analysis.

However, without serious improvements in the clarity, I do not believe it meets the quality criteria for publishing.

---

> ### Author Response · Authors · 2022-11-15
> **Author response to Reviewer 1W36**
>
> **1. Clarity**
>
> We appreciate the reviewer for the valuable advice. **We carefully revised the entire manuscript and included an additional diagram (Fig 1(a)) to provide better clarity.** Below, we provided the corrections and further explanations for the examples pointed out by the reviewer. **The other revisions are highlighted in Orange in the manuscript.**
>
> **List of Revisions**
> * **"The compatibility with the global basis represents how well the (optimal) global basis can work on the target latent space"**
>   * In this regard, the global-basis-compatibility represents how well the optimal global basis can work on the target latent space. Specifically, the global-basis-compatibility is defined as the quality of image traversal along the optimal global basis.
>
> * **"Our findings pave the way towards an unsupervised selection of globally disentangled latent space among the intermediate latent spaces in a GAN." in Abstract**
>   * Our work is the first step towards selecting the most disentangled latent space among various latent spaces in a GAN without attribute labels.
>
> * **"the number of semantically disentangled local perturbation from w" at Start of Sec 3**
>   * the number of local semantic perturbations from w
>
> * **"Because this optimal global basis is also the local disentangled perturbations at all latent variables" in Distortion and Global Disentanglement part of Sec 4**
>   * [Revised the two adjacent sentences together.] Then, there exists an optimal global basis of $\mathcal{M}$, where each basis vector corresponds to an image attribute on the entire $\mathcal{M}$. By definition, this optimal global basis is the local basis at all latent variables. Assuming that LB finds the local basis [1], each global basis vector would correspond to one LB vector at each latent variable.
>     * [Change of Notation in Response] Following Reviewer STT2's advice, we changed the notation for the learned latent manifold from $\mathcal{W}$ to $\mathcal{M}$.
>
> **List of Additional Explanations**
>
> * **Meaning of Fig 9**
>   * The goal of subspace traversal in Fig 9 is to compare the image fidelity under global basis traversal. We conducted this experiment to visually support the correlation between Distortion and the global-basis-compatibility demonstrated in Fig 7. Therefore, we compared the global basis traversal in the highest Distortion layer (Layer 3), smallest Distortion layer (Layer 7), and the renowned $\mathcal{W}$-space (Layer 8). The result demonstrates that the smallest Distortion layer indeed provides better image fidelity under the global basis traversal.
> * **"The global basis proposed in GANSpace is PCA components of latent variable samples"**
>   * As described in the Related Works Section, the definition of GANSpace is the PCA components of latent variable samples. We added a citation of GANSpace for a better presentation.
> ---
>
> **2. Comparison to [1]**
>
> We appreciate the reviewer for the valuable advice. **We clarified the contributions of this paper compared to [1] in "Local Basis" part of Related works. The main contributions are i) finding the number of semantic perturbations by local dimension estimation and ii) derivation of the global disentanglement metric.**
>
> i) [1] demonstrated that Local Basis serves as the local semantic perturbations. However, [1] did not provide an estimate of the number of these semantic perturbations. A manual inspection is required for each perturbation to check whether it is meaningful. For example, in $\mathcal{W}$-space in StyleGAN2, Local Basis provides 512 candidates of semantic perturbations. Our local dimension estimate can refine these candidates into about 40 candidates.
>
> ii) The global disentanglement metric can be derived because of our local dimension estimate. Without dimension estimate, the Grassmannian metric analysis in [1] relies on qualitative observation of the metric results for various subspace dimensions. Our local dimension estimation scheme provides a method for choosing a meaningful subspace dimension. This leads to the derivation of a layer-wise quantitative score. This derivation is important because it enables the comparison between latent spaces. Moreover, we conducted experiments on various StyleGAN models to verify that the proposed metric highly correlates with the global-basis-compatibility and supervised disentanglement score.
>
> ---
>
> **3. Reproducibility**
>
> We reorganized the code submitted in the supplementary material to provide better reproducibility.
>
> ---
>
> We hope that our response can address your main concerns. If so, we would like to kindly ask the reviewer to consider raising the score accordingly.
>
> **References**
> [1] Jaewoong Choi, Junho Lee, Changyeon Yoon, Jung Ho Park, Geonho Hwang, and Myungjoo Kang. Do not escape from the manifold: Discovering the local coordinates on the latent space of GANs. In International Conference on Learning Representations, 2022.

---

> > ### Comment · Reviewer_1W36 · 2022-11-27
> > **Concerns addressed in substance but clarity issues remaining**
> >
> > I thank the authors for the reply and improvements to the paper. I re-read the manuscript with a genuine desire to raise my score to acceptance by virtue of the paper's apparent substance value. Unfortunately, I remain surprised that vague statements remain even in the highlighted (revised) parts.
> >
> > Some examples of unclear phrasing: "The local intrinsic dimension is the number of dimensions..." , "how well the optimal global basis can work on the target latent space", or "Choi et al. (2022b) presents the candidates as much as the ambient dimension."
> >
> > I increased my score but subjectively I still consider the clarity of presentation to be, at best, borderline.

---

> > > ### Author Response · Authors · 2022-12-01
> > > **Thank you for the response**
> > >
> > > We appreciate the reviewer for spending time and effort to read our manuscript. Also, we are sorry for the vague statements contained in our revised manuscript. We clarified those examples further and provided the revised sentences below. We would incorporate these clarifications into the manuscript.
> > >
> > > **List of Revisions**
> > >
> > >  * The local intrinsic dimension is the number of dimensions required to properly approximate the latent space locally (Fig 1a). We discover this intrinsic dimension by estimating the robust rank of Jacobian of the subnetwork.
> > >    * Geometrically, the local dimension at $\mathbf{w} \in \mathcal{M}$ of the manifold $\mathcal{M}$ is the manifold dimension in the $\epsilon$-neighborhood of $\mathbf{w}$, i.e., the dimension of Euclidean space $\mathbb{R}^{k}$ whose open set is homeomorphic to the $\epsilon$-neighborhood of $\mathbf{w}$. In practice, the learned latent space exhibits some "noise". This noise makes it impossible to satisfy the strict bijective condition. Therefore, we define the local intrinsic dimension of learned latent space as the local dimension of "denoised" latent space. We discover this intrinsic dimension by estimating the noise-robust rank of subnetwork Jacobian.
> > >
> > >  * Here, the global-basis-compatibility means how well the optimal global basis can work on the target latent space.
> > >    * Here, the global-basis-compatibility of a latent space means the upper-bound on performance that the optimal global basis can achieve. In this paper, we adopted image fidelity as a performance measure (Sec 4).
> > >
> > >  * In this regard, the global-basis-compatibility represents how well the optimal global basis can work on the target latent space. Specifically, the global-basis-compatibility is defined as the quality of image traversal along the optimal global basis.
> > >
> > >    * In this regard, the global-basis-compatibility of a latent space refers to the highest performance that the optimal global basis can reach on it. Specifically, we chose the image quality of latent traversal along the optimal global basis as a measure of the global-basis-compatibility.
> > >
> > > * Since LB is defined as singular vectors, Choi et al. (2022b) presents the candidates as much as the ambient dimension.
> > >
> > >    * Since LB is defined as singular vectors, Choi et al. (2022b) presents the candidates as much as the minimum of $d_{\mathcal{Z}}$ and $d_{\mathcal{M}}$, e.g., 512 for $\mathcal{W}$-space in StyleGANs.

---

### Official Review · Reviewer_STT2 · 2022-10-26

**Confidence:** 3
**Correctness:** 4
**Technical Novelty And Significance:** 4
**Empirical Novelty And Significance:** 3
**Recommendation:** 5

**Clarity, Quality, Novelty And Reproducibility:**

The paper is somewhat hard to follow. For example the following sentence "The local intrinsic dimension is the dimension of approximating submanifold that well describes the latent space locally." is self referential. The paper could use another round of edits for clarity/being concise. The paper jumps directly into somewhat complex math that is not often found in GAN papers, a more gentle introduction to Local Basis (for example), would add to the clarity.

**Strength And Weaknesses:**

Strengths:
- The method for analyzing the mapping network seems novel.
- The paper provides a rigorous detailing of the pseudorank algorithm where its conclusion leads naturally to the Distortion metric.
- The proposed distortion metric is a an interesting and novel analysis tool for understanding GANs


Weaknesses:
- The evaluation could be more comprehensive. I'd like to see the distortion metric results on a variety of Karras et al.'s pretrained GANs versus FID.
- The lack of generated images is a little concerning. Figure 9 has some examples, but I am not quite sure what I am looking at. What attributes are being shown exactly? When the image travels from row 1 to row 5, I'm not sure that's a semantically meaningful attribute. The multi-dimensional travel is non-standard, maybe the center image could be highlighted in some what to indicate that it's the starting point. I'd also like to see examples from non-human faces GANs

Minor:
- I think a small diagram of a styleGAN to show the mapping network is where all the work is done may be illuminating for first time readers, as "intermediate layers of styleGAN" typically refer to the layers of the synthesis network.
- The notation abuse made my initial - skim a bit confusing. Perhaps calling it W- space (as opposed to W+ space) would be sufficient.

**Summary Of The Paper:**

This paper proposes a method for estimating local intrinsic dimensions in the intermediate layers of pretrained GANs, specifically with the mapping network of a stylegan. It also introduces a metric based of the method, which is used for analysis and comparison against similar metrics.

**Summary Of The Review:**

The paper is the first of its kind to use the intermediate layers of a stylegan mapping network to understand latent semantics. While the qualitative examples could use some work and while paper can, at times, be hard to follow I think the proposed analysis and metric are of interest to the broader community. Therefore I recommend weak accept.

Post-rebuttal:
I must admit, during my initial review I was uncertain about this paper. I believed my confusion regarding various mathematical concepts was due to my own lack of knowledge in this area. However, upon seeing the discussion with other reviews, it seems my confusion about clarity is not just related to myself. I do believe the content of the paper may be interesting to the community, but given the state of the writing I believe other readers will have as much of a difficulty as I did (even with the updates). I, unfortunately, must lower my score to weak reject.

---

> ### Author Response · Authors · 2022-11-15
> **Author response to Reviewer STT2**
>
> **1. The evaluation could be more comprehensive. I'd like to see the distortion metric results on a variety of Karras et al.'s pretrained GANs versus FID.**
>
> We appreciate the reviewer for the valuable advice. Following the reviewer's advice, **we conducted additional experiments on LSUN Cat and Horse datasets.** The experimental results are presented at Fig 19 and 20 in the appendix. In both datasets, Distortion metric shows similar robust correlations to FID under four preprocessing hyperparameters $\theta_{pre}$.
>
> > Added FID Correlation results and highlighted in Brown in Fig 19, 20
> ---
> **2. The lack of generated images is a little concerning. Figure 9 has some examples, but I am not quite sure what I am looking at. What attributes are being shown exactly? When the image travels from row 1 to row 5, I'm not sure that's a semantically meaningful attribute. The multi-dimensional travel is non-standard, maybe the center image could be highlighted in some what to indicate that it's the starting point. I'd also like to see examples from non-human faces GANs**
>
> We appreciate the reviewer for the constructive advice. **We highlighted the starting image in the center with a blue square and the image failure with a yellow square. In addition, more image examples are added to the appendix (LSUN Cat: Fig 28, LSUN Church: Fig 29, LSUN Horse: Fig 30, LSUN Car: Fig 31)**. The goal of subspace traversal in Fig 9 is to compare the image fidelity under global basis traversal. Here, this image fidelity shows the global-basis-compatibility of each latent space. We chose image fidelity as a comparison criterion between image samples because it is more objective than semantic factorization. We agree with the reviewer that multi-dimensional traversal is rather non-standard than linear traversal. However, we chose the multi-dimensional traversal because it is more challenging as a task itself without relying on the perturbation intensity.
>
> > Added additional traversal images and highlighted in Brown in Fig 28, 29, 30, 31
> ---
> **3. Minor issues**
>
> Thank you for the detailed advice.
>
> * We added a small diagram of StyleGAN to the appendix, highlighting the mapping network where our evaluation is conducted. Also, we included an explanation about where the architecture diagram is provided in Section 3.3.
> > Added architecture diagram of StyleGAN at Fig 11
> * To avoid confusion, we changed the notation from $\mathcal{W}$-space to $\mathcal{M}$-space.
> ---
> **4. Clarity**
>
> Thank you for the thoughtful comment. We carefully revised the entire manuscript to provide better clarity. The revisions are highlighted in Orange in the manuscript. Below, we provided the corrections and further explanations for the examples pointed out by the reviewer.
>
> * **"The local intrinsic dimension is the dimension of approximating submanifold that well describes the latent space locally." at the top of Page 2**
>   * The local intrinsic dimension is the number of dimensions required to properly approximate the latent space locally.
> * **Gentle introduction to Local Basis**
>   * We added more detailed explanations of Local Basis in Related Works Section and highlighted them in Brown. Also, we included the concept diagram of the local dimension estimation at Fig 1(a) which can provide better clarity to our work and Local Basis.
> ---
> We hope that our response can address your main concerns. If so, we would like to kindly ask the reviewer to consider raising the score accordingly.

---

> > ### Author Response · Authors · 2022-12-01
> > **Thank you for the response.**
> >
> > We appreciate the reviewer for spending time and effort to re-evaluate our manuscript. Before changing the score, we hope the reviewer to consider **our response to the initial review** and **other assessments on clarity** (good clarity in RU6v, easy to follow in dnDf). Below, we provided **further clarified sentences**, which would be incorporated into the manuscript. We hope these revisions to be helpful in addressing the reviewer's concern about clarity.
> >
> > **List of Revisions**
> >
> >  * The local intrinsic dimension is the number of dimensions required to properly approximate the latent space locally (Fig 1a). We discover this intrinsic dimension by estimating the robust rank of Jacobian of the subnetwork.
> >    * Geometrically, the local dimension at $\mathbf{w} \in \mathcal{M}$ of the manifold $\mathcal{M}$ is the manifold dimension in the $\epsilon$-neighborhood of $\mathbf{w}$, i.e., the dimension of Euclidean space $\mathbb{R}^{k}$ whose open set is homeomorphic to the $\epsilon$-neighborhood of $\mathbf{w}$. In practice, the learned latent space exhibits some "noise". This noise makes it impossible to satisfy the strict bijective condition. Therefore, we define the local intrinsic dimension of learned latent space as the local dimension of "denoised" latent space. We discover this intrinsic dimension by estimating the noise-robust rank of subnetwork Jacobian.
> >
> >  * Here, the global-basis-compatibility means how well the optimal global basis can work on the target latent space.
> >    * Here, the global-basis-compatibility of a latent space means the upper-bound on performance that the optimal global basis can achieve. In this paper, we adopted image fidelity as a performance measure (Sec 4).
> >
> >  * In this regard, the global-basis-compatibility represents how well the optimal global basis can work on the target latent space. Specifically, the global-basis-compatibility is defined as the quality of image traversal along the optimal global basis.
> >
> >    * In this regard, the global-basis-compatibility of a latent space refers to the highest performance that the optimal global basis can reach on it. Specifically, we chose the image quality of latent traversal along the optimal global basis as a measure of the global-basis-compatibility.
> >
> > * Since LB is defined as singular vectors, Choi et al. (2022b) presents the candidates as much as the ambient dimension.
> >
> >    * Since LB is defined as singular vectors, Choi et al. (2022b) presents the candidates as much as the minimum of $d_{\mathcal{Z}}$ and $d_{\mathcal{M}}$, e.g., 512 for $\mathcal{W}$-space in StyleGANs.

---

### Official Review · Reviewer_dnDf · 2022-10-27

**Confidence:** 4
**Correctness:** 3
**Technical Novelty And Significance:** 3
**Empirical Novelty And Significance:** 3
**Recommendation:** 6

**Clarity, Quality, Novelty And Reproducibility:**

This work is well written, experimentally supported and clear to follow. It is originally built over an existing method, and proposes new solutions to the latent space manipulation problem of GANs. The codes are shared.I ﬁnd most of the intuitive and theoretical parts plausible. The general structure of the paper is consistent. Even though I ﬁnd the work worthy, especially the Distortion metric, I still have minor doubts about the local intrinsic dimension estimation in terms of its novelty and necessity for the literature.


**Strength And Weaknesses:**

Strengths:

1. The ﬂow of the paper is well designed and easy to follow.

2. Theoretical parts seem plausible. Most of the methodologies are also explained intuitively. Experimental results support both the paper’s hypothesis and the previous literature.

3. The proposed metric Distortion might be useful to validate the global disentanglement of a learnt latent space for the upcoming works.

 Weaknesses:

1. At the end of “Understanding Latent Semantics” part in Related Work, the authors state that the global methods showed promising results, but also said that they were successful in a limited area. This sentence deserves further explanation.


2. In “Sparsity Constraint” part, the authors compare their method with the LowRankGAN’s rank estimation method, and state that “saturation should occur to ﬁnd an intrinsic rank”. I’m not convinced enough with this explanation. I think a clearer explanation should be made.


3. In Figure 5(b) and and Figure 13(b-e), traversals along the (d-1)-th and d-th axis where d indicates the estimated dimension are shown. It would be good to see the visual impacts of traversing along the axis greater than d since the authors present d as an upper bound on the number of perturbations. Would it display random/noisy traversals, as those dimensions are redundant ?

**Summary Of The Paper:**

In this paper, the authors propose an unsupervised, geometrically aware local intrinsic dimension estimation algorithm for latent space manipulation of GANs, along with a metric called “Distortion” as a global disentanglement score which is constructed with the help of the local intrinsic dimension. They build their work on a previously proposed method called “Local Basis” which gives local, semantically disentangled directions for style space W of StyleGANs. While these direction vectors are obtained from the solution of low-rank approximation to the linear map df_z between the tangent spaces of Z and W, they are already shown to demonstrate meaningful, stable manipulations. However, a lower dimensional approximation might be possible. Therefore, the authors of this paper presents a method to estimate this intrinsic dimension. As the vectors of Local Basis are obtained from the Jacobian, the authors ﬁrst eliminate the singular values of the Jacobian which cause overestimation of the intrinsic rank. Then, they apply Pseudorank algorithm to the Jacobian to diﬀerentiate meaningful components from the noisy components where the rank of the Jacobian indicates this diﬀerence. They experimentally validate their estimations of the intermediate layers in the mapping network and visually show manipulation results in the image space. Besides, they calculate a global disentanglement score which is related to the inconsistency of intrinsic tangent spaces. They experimentally show the correlation of this metric with the global disentanglement.

**Summary Of The Review:**

I ﬁnd most of the intuitive and theoretical parts plausible. The general structure of the paper is consistent. Even though I ﬁnd the work worthy, especially the Distortion metric, I still have minor doubts about the local intrinsic dimension estimation in terms of its novelty and necessity for the literature.

---

> ### Author Response · Authors · 2022-11-15
> **Author response to Reviewer dnDf**
>
> **1. At the end of “Understanding Latent Semantics” part in Related Work, the authors state that the global methods showed promising results, but also said that they were successful in a limited area. This sentence deserves further explanation.**
>
> We thank the reviewer for the valuable advice. We added additional explanations to the end of “Understanding Latent Semantics” part in Related Work. The revision is as follows:
>
> -  These global methods showed promising results, but they were successful in a limited area. Depending on the sampled latent variables, these methods exhibited limited semantic factorization and sharp degradation of image fidelity [1,2].
>
> > Added additional explanations and highlighted in Blue to the end of “Understanding Latent Semantics” part in Sec 2
> ---
> **2. In “Sparsity Constraint” part, the authors compare their method with the LowRankGAN’s rank estimation method, and state that “saturation should occur to ﬁnd an intrinsic rank”. I’m not convinced enough with this explanation. I think a clearer explanation should be made.**
>
> We thank the reviewer for the careful advice. We clarified the statement in the “Sparsity Constraint” part. The revision is as follows:
>
> - We consider that the rank saturation should occur if this assumption is adequate for finding an *intrinsic* rank because it implies regularization robustness.
>
> > Added additional explanations and highlighted in Blue to the end of “Sparsity Constraint” part in Sec 3.2
> ---
> **3. In Figure 5(b) and and Figure 13(b-e), traversals along the (d-1)-th and d-th axis where d indicates the estimated dimension are shown. It would be good to see the visual impacts of traversing along the axis greater than d since the authors present d as an upper bound on the number of perturbations. Would it display random/noisy traversals, as those dimensions are redundant?**
>
> We appreciate the reviewer for the insightful advice. **We added additional image traversal results along the axis $i> d$ in Fig 24 in the appendix, where d indicates the estimated dimension. When we traverse along the axis $i \gg d$, the images show much smaller variation than the axis $i \leq d$ or overlap with each other.** This result supports our dimension estimation in two aspects. First, we think these minor image variations on the axis $i \gg d$ validate our approach, i.e., finding the intrinsic dimension through denoising of Jacobian. During the training process, the margin of the latent manifold along axis $i \gg d$ is too thin for the synthesis network to represent the meaning semantics. Therefore, the synthesis network makes a "safe choice" of interpreting these axes as noise, so these axes show minor variations. Second, there is not much difference between the axes $i \gg d$. For example, axis 100 and axis 500 exhibit a similar degree of image variations. We guess this is because these axes are characterized only by the orthogonal complement of intrinsic tangent spaces. Hence, the ordering between axis 100 and axis 500 is not meaningful. Therefore, it is natural to investigate the intrinsic tangent spaces.
>
> > Added additional Linear Traversal and highlighted in Blue at Fig 24
> ---
> **4. I still have minor doubts about the local intrinsic dimension estimation in terms of its novelty and necessity for the literature.**
>
> We believe the local dimension estimation scheme has the potential to be applied to various models. In particular, **this scheme can be used to locally estimate the learned feature space for more general input, e.g., image, with the proper assumption.** When using the notation of Local Basis paragraph in Sec 2, the required assumption is that $\epsilon$-perturbation of input image is still valid to guarantee that domain of $df_{\mathbf{z}}$ is $\mathbb{R}^{d_{\mathcal{Z}}}$, i.e., $T_{\mathbf{z}}\mathcal{Z}=\mathbb{R}^{d_{\mathcal{Z}}}$. This local estimation of feature space can be applied to diverse tasks. For example, the adversarial robustness of the classifier can be analyzed by projecting the adversarial noise onto the estimated feature space. Also, manifold learning can be applied to this local estimation of feature space. This local estimation can be beneficial for manifold learning because it can provide a lower-dimensional structure regardless of large ambient space. **We added this discussion to Conclusion as future work.**
>
> > Added discussion to Conclusion and highlighted in Red
> ---
> We hope that our response can address your main concerns. If so, we would like to kindly ask the reviewer to consider raising the score accordingly.
>
> **References**
> [1] Jaewoong Choi, Junho Lee, Changyeon Yoon, Jung Ho Park, Geonho Hwang, and Myungjoo Kang. Do not escape from the manifold: Discovering the local coordinates on the latent space of GANs. In ICLR, 2022.
> [2] Jaewoong Choi, Geonho Hwang, Hyunsoo Cho, and Myungjoo Kang. Finding the global semantic representation in gan through frechet mean. arXiv preprint arXiv:2210.05509, 2022.

---

> > ### Author Response · Authors · 2022-12-08
> > **Thank you for the thoughtful feedback.**
> >
> > Thank you for the thoughtful feedback.
> >
> > We hope our responses were helpful in addressing the reviewer's concerns.
> >
> > If our responses were helpful, we would like to gently ask the reviewer to consider re-evaluating our manuscript.
> >
> > If you have any further concerns, feel free to leave a comment. We are waiting for the reviewer's response.

---

### Official Review · Reviewer_RU6v · 2022-11-02

**Confidence:** 4
**Correctness:** 4
**Technical Novelty And Significance:** 2
**Empirical Novelty And Significance:** 2
**Recommendation:** 6

**Clarity, Quality, Novelty And Reproducibility:**

The paper is in good clarity and quality.

The novelty is good, however, I doubt if this is an interesting topic for the field now.

**Strength And Weaknesses:**

The strength of paper:

The paper look into style-GAN and explore a new metric named as Disentanglement Score. This is a groundbreaking method.
Also this paper illustrate a good method to evaluate GAN by a deep look into latent space.


**Summary Of The Paper:**

In this paper, the authors proposed a local intrinsic dimension estimation algorithm for the intermediate latent
space in a pre-trained GAN. Using this algorithm, we analyzed the intermediate layers in the mapping
network of StyleGANs on various datasets.

The paper is very straight-forward. It raised a concept to evaluate GAN by exploring the latent space of Generative model.

**Summary Of The Review:**

I think this is a ground breaking paper for evaluating and improving the robustness of GAN. The authors have applied strict method to evaluate that. The only concern is that the authors should show more to illustrate if this method can contribute to the field now.

---

> ### Author Response · Authors · 2022-11-15
> **Author response to Reviewer RU6v**
>
> **1. Contribution to the field**
>
> Thank you for the thoughtful advice. We are encouraged that the reviewer found our manuscript groundbreaking. We believe our method can contribute to the field by **providing a criterion for choosing the most disentangled latent space without attribute labels.** Moreover, the local dimension estimation scheme has the potential to be applied to various models. In particular, **this scheme can be used to locally estimate the learned feature space for more general input, e.g., image, with the proper assumption.** When using the notation of Local Basis paragraph in Sec 2, the required assumption is that $\epsilon$-perturbation of input image is still valid to guarantee that domain of $df_{\mathbf{z}}$ is $\mathbb{R}^{d_{\mathcal{Z}}}$, i.e., $T_{\mathbf{z}}\mathcal{Z}=\mathbb{R}^{d_{\mathcal{Z}}}$. This local estimation of feature space can be applied to diverse tasks. For example, the adversarial robustness of the classifier can be analyzed by projecting the adversarial noise onto the estimated feature space. Also, manifold learning can be applied to this local estimation of feature space. This local estimation can be beneficial for manifold learning because it can provide a lower-dimensional structure regardless of large ambient space. **We added this discussion to Conclusion as future work.**
>
> > Added discussion to Conclusion and highlighted in Red
>
> We hope that our response can address your main concerns. If so, we would like to kindly ask the reviewer to consider raising the score accordingly.

---

> > ### Author Response · Authors · 2022-12-08
> > **Thank you for the thoughtful feedback.**
> >
> > Thank you for the thoughtful feedback.
> >
> > We hope our responses were helpful in addressing the reviewer's concerns.
> >
> > If our responses were helpful, we would like to gently ask the reviewer to consider re-evaluating our manuscript.
> >
> > If you have any further concerns, feel free to leave a comment. We are waiting for the reviewer's response.

---

### Author Response · Authors · 2022-11-15
**Author response to all reviewers**

We deeply thank the reviewers for spending time reading our manuscript carefully and providing thoughtful feedback. We think that the reviewers raised several valuable questions, and answering those questions has significantly improved our work. Below we address specific questions and comments to each reviewer. We highlighted the corresponding revisions in the manuscript in Red for Reviewer RU6v, Blue for Reviewer dnDf, Brown for Reviewer STT2, and Orange for Reviewer 1W36.

---

### Decision · Program_Chairs · 2023-01-20

**Decision:**

Reject

**Justification For Why Not Higher Score:**

The empirical evaluation was weak, with limited insight into whether the latent space traversals correspond to anything semantically meaningful.

**Justification For Why Not Lower Score:**

N/A

**Metareview: Summary, Strengths And Weaknesses:**

The authors propose a method for estimating the number of disentangled local perturbations. Strengths include a relatively well motivated method, and the proposal of a distortion metric that correlates well with supervised disentanglement score. The major weakness of the paper, as noted by the reviewers, is the limited empirical evaluation. It is difficult to know whether the latent space traversals correspond to anything semantically meaningful, and whether these are better than predecessors. Furthermore, it's applicability to only StyleGAN models limits its applicability. Finally, as some reviewers noted, the presentation could be clearer